



# Pragmatic Solvers for 3D Stokes and Elasticity Problems with Heterogeneous Coefficients: Evaluating Modern Incomplete $LDL^T$ Preconditioners

Patrick Sanan[1], Dave A. May[2], Matthias Bollhöfer[3], and Olaf Schenk[4]

[1]Department of Earth Sciences, ETH Zurich, Sonneggstrasse 5, Zürich 8092, Switzerland
[2]Department of Earth Sciences, University of Oxford, South Parks Road, Oxford OX1 3AN, United Kingdom
[3]Institute for Computational Mathematics, TU Braunschweig, D-38106 Braunschweig, Germany
[4]Advanced Computing Laboratory, Università della Svizzera italiana, Via Buffi 6, Lugano 6900, Switzerland

**Correspondence:** Patrick Sanan (patrick.sanan@erdw.ethz.ch)

**Abstract.** The need to solve large saddle point systems within computational Earth sciences is ubiquitous. Physical processes giving rise to these systems include porous flow (the Darcy equations), poroelasticity, elastostatics, and highly viscous flows (the Stokes equations). The numerical solution of saddle point systems is non-trivial since the operators are indefinite.

Primary tools to solve such systems are direct solution methods (exact triangular factorization) or Approximate Block Factorization (ABF) preconditioners. While ABF solvers have emerged as the state-of-the-art scalable option, they are invasive solvers requiring splitting of pressure and velocity degrees of freedom, a multigrid hierarchy with tuned transfer operators and smoothers, machinery to construct complex Schur complement preconditioners, and the expertise to select appropriate parameters for a given coefficient regime – they are far from being "black box" solvers. Modern direct solvers, which robustly produce solutions to almost any system, do so at the cost of rapidly growing time and memory requirements for large problems, especially in 3D. Incomplete $LDL^T$ (ILDL) factorizations, with symmetric maximum weighted matching preprocessing, used as preconditioners for Krylov (iterative) methods, have emerged as an efficient means to solve indefinite systems. These methods have been developed within the numerical linear algebra community but have yet to become widely used in non-trivial applications, despite their practical potential; they can be used whenever a direct solver can, only requiring an assembled operator, yet can offer comparable or superior to performance, with the added benefit of having a much lower memory footprint. In comparison to ABF solvers, they only require the specification of a drop tolerance and thus provide an easy-to-use addition to the solver toolkit for practitioners.

Here, we present solver experiments employing incomplete $LDL^T$ factorization with symmetric maximum weighted matching preprocessing to precondition operators, and compare these to direct solvers and ABF-preconditioned iterative solves. To ensure the comparison study is meaningful for Earth scientists, we utilize matrices arising from two prototypical problems, namely Stokes flow and quasi-static (linear) elasticity, discretized using standard mixed finite element spaces. Our test suite targets problems with large coefficient discontinuities across non-grid-aligned interfaces, which represent a common, challenging-for-solvers, scenario in Earth science applications. Our results show: (i) as the coefficient structure is made increasingly challenging (high contrast, complex topology), the ABF solver can break down, becoming less efficient than the



ILDL solver before breaking down entirely; (ii) ILDL is robust, with a time-to-solution that is largely independent of the co-
efficient topology and mildly dependent on the coefficient contrast; (iii) the time-to-solution obtained using ILDL is typically
faster than that obtained from a direct solve, beyond $10^5$ unknowns; (iv) ILDL always uses less memory than a direct solve.

## 1  Introduction

Saddle point systems frequently arise in the context of constrained minimization problems. Many physical processes relevant to
the Earth sciences fall within such a minimization framework. Possibly the most widely used relates to the variational statement
which seeks to constrain a vector field (e.g. displacement / velocity) to be divergence free. Such formulations naturally introduce
a pressure-like (scalar) variable to constrain the displacement / flow (vector) field. Specific examples include: mixed Darcy
problems involving the unknowns Darcy flux and saturation pressure (porous flow, groundwater flow, oil and gas reservoirs);
poroelasticity (geomechanics, reservoirs engineering, bore hole stability); compressible / incompressible quasi-static linear
elasticity (crustal deformation targeting on inter-seismic periods); and incompressible viscous flow (dynamics of the mantle,
lithosphere, glaciers, ice sheets). Other relevant (but more generic) applications giving rise to discrete problems of saddle
point type include PDE-constrained optimization, weak imposition of boundary conditions (e.g. contact, fault constitutive
behaviour) and matching conditions between different model domains (e.g. Beavers-Joseph matching conditions between fluid-
solid regions).

Solution techniques for saddle point systems have been extensively studied (Benzi et al., 2005; Benzi and Wathen, 2008;
Loghin and Wathen, 2004). Nevertheless, saddle point systems have a reputation for being challenging to solve - this is to a
large part attributed to the fact that the discrete problem (e.g. the matrix), while often symmetric, is *indefinite*; this structure
precludes the use of many standard approaches (like classical multigrid).

Saddle point problems too large to be practically solved via sparse direct solution techniques (e.g. $LU$ or $LDL^T$ factoriza-
tion) are deemed to require highly-specialized solvers. Indeed, if algorithmically optimal methods are sought, this opinion is
justified. As such, the development of highly scalable, optimal preconditioners for the solution of large-scale variable viscosity
Stokes systems arising in ice sheet modeling (Isaac et al., 2015) and geodynamics (May et al., 2014; Rudi et al., 2015) is
mature.

However, the *practical* usage of saddle point solvers does not always favor the above agenda. Maximum problem sizes of
interest are typically fixed or modest (e.g. $< O(10^8)$ DOFs); algorithmic or parallel scalability may not be to be prized to the
exclusion of all else; time-to-solution(s) may not dominate the time required to set up and tune (by hand) a specialized solver;
or computational resources available may be modest (e.g. single compute node with 100 GB RAM and unlimited walltime,
or few low memory compute nodes with $\sim 400$ cores with walltime restricted to $< 24$ hrs). Specialized, optimal solvers often
lack robustness as problem parameters or problem types are varied, though progress is being made in this regard (e.g. Rudi
et al., 2017).

Solvers can trade some algorithmic performance for robustness and/or ease of use. A striking example is the persistent use of
incomplete LU (ILU) preconditioning (e.g. Wathen (2015) §6.1, Benzi (2002) §3). With the help of an easy-to-apply-and-store



approximate inverse to the system matrix, obtained by discarding terms from a full factorization (as used in a direct solver), the solution may be iteratively updated until it approaches the solution. While ILU-preconditioned Krylov methods are neither algorithmically scalable nor completely robust, the method is ubiquitous for several reasons.

1. *Only an assembled operator is required.* Many efficient preconditioners require auxiliary information, typically concerning the physical domain of an underlying PDE. For example, multigrid solvers require a hierarchy of grids and transfer operators and FETI-DP (Farhat et al., 2001) methods require access to finite element subdomains. However, purely algebraic solvers, which only require an assembled operator (matrix), are always in demand, as they can be applied much more broadly and allow for greater ease of algorithmic experimentation.

2. *Reasonable performance is observed for a large class of relevant problems.* Some variants of ILU preconditioning reduce the condition number of a standard second-order elliptic PDE, discretized with finite differences or finite elements, from $O(h^2)$ to $O(h)$ (Benzi, 2002) As such, the preconditioner has been used in many applications, alone or as a subdomain preconditioner for a block-Jacobi preconditioner, for example in subsurface flow (Mills et al., 2007).

    3. *The methods are tunable, with a small number of parameters.* ILU-preconditioned Krylov methods typically only expose
a small number of parameters to the user, depending on the variant employed (Saad, 2003, Ch. 10). For instance, in ILU(k) methods, the nonzero entries of the factors are restricted to those of $A^k$, one may drop any entries below a given threshold, or use more complicated approaches such as ILUT (Saad, 1994). Multi-level inverse-based dropping strategies (Bollhöfer and Saad, 2006) accept a drop tolerance for factorization as well as a potentially different drop tolerance.

    Denser factors typically produce a better approximate inverse, which typically makes for a better preconditioner. Thus,
a simple trade-off exists between iteration counts and memory usage. This is in contrast to direct solvers, which do not offer the user any control over memory usage and which always (over-)solve to machine precision.

    Often, an additional choice is available of an ordering strategy such as approximate minimum degree ordering, nested dissection, reverse Cuthill-McKee (RCM), or others. Finally, a choice must be made of the Krylov method itself; this study uses right-preconditioned GMRES or FGMRES, but this is rarely a critical choice and many options are usually
available, as Krylov methods are far simpler to implement than direct solvers or incomplete factorization preconditioners.

    4. *Tools are widely available in software.* Because of the conveniences described in points 1 and 3, ILU preconditioners have made their way into numerous software packages, including MATLAB (MATLAB, 2019), ILUPACK (Bollhöfer and Saad, 2006), UMFPACK (Davis, 2004), WSMP (Gupta et al., 1997), PETSc (Balay et al., 2019a, b), TRILINOS (Heroux et al., 2005), and many others, as they often provide a reasonable default preconditioner.

5. *The method is well-discussed in the practical literature.* Potential users are likely to have access to a performance study which includes the effect of ILU preconditioning, with documentation of the parameter choices employed. A valid criticism of ILU preconditioning is that part of its popularity is simply due to its heritage; it was studied early and hence there is robust literature which directly demonstrates its use.



Incomplete $LDL^T$ (ILDL) preconditioners arise as incomplete versions of direct solvers for indefinite systems, which use
the factorization $A = LDL^T$, where $L$ is lower-triangular and $D$ is block diagonal. Here the term "incomplete" implies that
factorization is only approximate. ILDL methods with symmetric maximum weighted-matching preprocessing (see Section 3)
have emerged in the numerical linear algebra community as an analogous method to ILU methods, but for symmetric indefinite
systems; they are more robust than ILU or previous incarnations of ILDL. Recently, comparison studies characterizing such
approaches have appeared: Hagemann and Schenk (2006) provide a study of the effectiveness of ILDL preconditioning with
respect to a matrix "zoo"; Greif et al. (2015) performed a similar study with their SYM-ILDL package; Multilevel ILDL pre-
conditioners with symmetric weighted matching preprocessing have been shown to be effective when applied to the Helmholtz
equation (Bollhöfer et al., 2009); Schenk et al. (2008) showed that a similar approach can also be used to effectively compute
a few interior eigenvalues of a large indefinite matrix arising from the Anderson model of localization. Recent work on sparse
inverse covariance matrix estimation highlights how ILDL preconditioners can be preferable to highly efficient direct solvers,
due to their much lower memory footprints (Bollhöfer et al., 2019a).

Despite these studies, the applicability of ILDL preconditioners is less well-known outside the numerical linear algebra
community. This is unfortunate, as these modern methods bring incomplete factorization approaches for indefinite symmetric
systems in line with other popular methods, in terms of robustness and ease of use. It should also be emphasized that points 1
and 3 hold for the ILDL method.

### 1.1 Motivations and Outline

This paper addresses points 2, 4, and the beginnings of 5 in the context of using ILDL preconditioners with symmetric max-
imum weighted matching ordering applied to saddle point systems arising from the spatial discretization of a class of PDEs
relevant to the solid Earth.

In contrast to the previous ILDL studies previous mentioned above, which mostly focus on the robustness of preconditioners
across a corpus of matrices representing individual instances of different applications, here we focus on a deeper examination
of a specific class of PDEs commonly used within the Earth sciences. Specifically, we wish to examine the saddle point
problems arising from stationary Stokes flow with highly heterogeneous viscosity structure, and systems arising from the static
linear elasticity, also with large coefficient jumps. Particular attention is paid to physical problems with challenging, non-grid-
aligned coefficient jumps, as these constitute some of the more challenging systems of interest within the Earth sciences. This
focus allows new insight into the effect of varying problem size and parameters (in particular, coefficient structure) on the
performance of solvers. We only examine the scenario in which a given linear system need only be solved for a single right
hand side. This is typical when solving nonlinear systems of equations (often within time-stepping algorithms) when a Jacobian
and residual are assembled and used to compute a step.

The saddle point operators arising from the Stokes and static elasticity (in mixed form) systems are presented in Section 2.
Section 3 describes the incomplete ILDL factorization preconditioner with maximum symmetric weighted matching prepro-
cessing. Section 4 describes an Approximate Block Factorization (ABF) preconditioner and a sparse direct solver, which serve
as representatives of the two classes of alternative approaches in common use and which we will compare ILDL against. Section





5 presents experiments which characterize the performance of these ILDL-preconditioners, direct and the ABF-preconditioned solves applied to the Stokes and elasticity problems for a variety of synthetic model configurations involving multiple inclu-
sions of differing material parameters (with respect to the surrounding medium).

## 2 Prototypical problems and saddle point systems

### 2.1 Stokes flow

Conservation of momentum and mass for an incompressible creeping fluid in a domain $\Omega$ with boundary $\partial\Omega$ are given by

$$-\nabla \cdot \tau + \nabla p = \rho\hat{g},$$
$$-\nabla \cdot u = 0, \tag{1}$$

where $u$ and $p$ are the velocity and pressure, respectively. The forcing term is associated with buoyancy variations; $\rho$ is a spatially-varying density and $\hat{g}$ is the gravity vector. For the isotropic media we consider here, the deviatoric stress $\tau$ is related to the strain rate $\dot{\varepsilon}[u]$ via

$$\tau = 2\eta\dot{\varepsilon}[u], \quad \dot{\varepsilon}[u] = \tfrac{1}{2}\left(\nabla u + (\nabla u)^T\right), \tag{2}$$

where $\eta$ is a spatially-varying effective shear viscosity. The system given by Eq. (1) is closed with appropriate boundary
conditions specified on the normal and tangential components of the velocity and stress ($\sigma$). In this work, we consider the following boundary conditions:

$$u \cdot n = 0, \quad t \cdot \tau \cdot n = 0 \qquad x \in \Gamma_D, \tag{3}$$
$$n \cdot \sigma \cdot n = 0, \quad t \cdot \sigma \cdot n = 0 \qquad x \in \Gamma_F, \tag{4}$$

where $\sigma = \tau - pI$ is the total stress, $n, t$ are the normal (outward pointing) and tangent vector to the boundary $\partial\Omega$, for which
$\Gamma_D \cap \Gamma_F = \emptyset$ and $\Gamma_D \cup \Gamma_F = \partial\Omega$.

### 2.1.1 Discrete problem

We use inf-sup stable mixed finite elements (FE) to obtain discrete solutions of Eq. (1). A full description of the variational (weak) problem associated with incompressible Stokes flow can be found in Elman et al. (2005). The discrete Stokes problem $\mathcal{A}$ is denoted by

$$\begin{bmatrix} K & B^T \\ B & 0 \end{bmatrix} \begin{bmatrix} u \\ p \end{bmatrix} = \begin{bmatrix} F \\ 0 \end{bmatrix}, \quad \text{or } \mathcal{A}v = \mathcal{F}, \tag{5}$$

where $K$ is the discrete gradient of the deviatoric stress tensor and $B, B^T$ are the discrete gradient and divergence operators respectively. We note that the FE discretization results in $K$ being symmetric positive definite, thus $\mathcal{A}$ is a symmetric, indefinite operator.



## 2.2 Static linear elasticity

The conservation of momentum for an elastic solid in static equilibrium in a domain $\Omega$ with boundary $\partial\Omega$ is given by

$$-\nabla \cdot \hat{\sigma} = \rho \hat{g}, \tag{6}$$

where $\rho$ and $\hat{g}$ were defined previously in Section 2.1, and $\hat{\sigma}$ is the total stress, which we assume to obey the linear, isotropic constitutive relation

$$\begin{aligned}
\hat{\sigma} &= 2\mu \varepsilon[u] + \lambda \operatorname{Tr}(\varepsilon[u])\, I \\
&= 2\mu \varepsilon[u] + \lambda \nabla \cdot u\, I,
\end{aligned} \tag{7}$$

where $u$ is the displacement and the (linear) strain tensor $\varepsilon[u]$ is given by

$$\varepsilon[u] = \tfrac{1}{2}\left(\nabla u + (\nabla u)^T\right) \tag{8}$$

and $\operatorname{Tr}(\cdot)$ denotes the trace operator. The particular form of the constitutive relationship adopted is defined in terms of the two *Lamé* parameters $(\lambda, \mu)$. The first Lamé parameter, $\lambda$, characterizes compressibility; as $\lambda \to \infty$, the material becomes incompressible. The second Lamé parameter, $\mu$, is equivalent to the shear modulus (often denoted $G$) and characterizes resistance to
shearing.

Describing materials which are incompressible in some or all of $\Omega$ is problematic with the formulation given in Eqs. (6) and (7) since the last term in Eq. (7) behaves like $\infty \times 0$ in the incompressible limit. The latter scenario is relevant when considering plasticity models, or the presence of fluids (e.g. within a crack along the subduction interface). Even when large but finite values for $\lambda$ (e.g. the equivalent Poisson ratio $\nu = \frac{\lambda}{2(\lambda+\mu)} > 0.49$) are used, "locking" may occur when using standard
finite difference, finite volume, or finite element spatial discretizations (Brezzi and Fortin, 1991), rendering the displacement solutions meaningless. The issues in the incompressible limit can be resolved by grouping the problematic terms into a new auxiliary pressure variable, $p = -\lambda \nabla \cdot u$. Then, if we decompose the stress as $\hat{\sigma} = \tau - pI$, Eqs. (6) and (7) can be cast as the following mixed $(u, p)$ problem (Brezzi and Fortin, 1991):

$$\begin{aligned}
-\nabla \cdot \tau + \nabla p &= \rho(x)\hat{g}, \\
-\nabla \cdot u - (1/\lambda)p &= 0,
\end{aligned} \tag{9}$$

with stress $\tau$ given by

$$\tau = 2\mu \varepsilon[u]. \tag{10}$$

One will observe that form is similar to the Stokes system described in Section 2.1; $u$ now represents displacements, strain is considered instead of strain rate, and pressure and divergence of $u$ are now related by the material parameter $\lambda$. This implies that the degree of coupling between $u$ and $p$ within the conservation of mass may be spatially variable as $\lambda$ need not be constant
throughout $\Omega$. One should also note that $\tau$ in Eq. (10) is only deviatoric in regions where $\lambda \to \infty$, cf. Stokes where $\tau$ is strictly





deviatoric everywhere in $\Omega$ since $\nabla \cdot v = 0$ is imposed throughout the domain. The boundary conditions given by Eqs. (3), (4) are valid for the mixed elasticity problem, with the velocity $v$ interchanged for the displacement $u$, and $\sigma$ interchanged with $\hat{\sigma}$. Additionally, since the stress decomposition in the mixed elasticity formulation is not strictly deviatoric-volumetric (e.g. volumetric terms still live within $\hat{\sigma}$), we replace $\tau$ in Eq. (3) with dev$(\hat{\sigma})$.

### 2.2.1 Discrete problem

The discrete mixed $(u, p)$ static elasticity problem $\mathcal{A}_{\mathrm{L}}$ is denoted by

$$\begin{bmatrix} K & B^T \\ B & -C \end{bmatrix} \begin{bmatrix} u \\ p \end{bmatrix} = \begin{bmatrix} F \\ 0 \end{bmatrix}, \quad \text{or } \mathcal{A}_{\mathrm{L}} v = \mathcal{F}, \tag{11}$$

where $K$ is the discrete gradient of the deviatoric stress tensor and $B, B^T$ are the discrete gradient and divergence operators, and $C$ is a $1/\lambda$-weighted discrete mass matrix. The FE discretization results in $K$ and $C$ being symmetric positive definite and, thus $\mathcal{A}_{\mathrm{L}}$ is again a symmetric, indefinite operator. To ensure the discrete problem is stable when the continuum is incompressible (or near to this limit), be it locally or globally (as determined by the value of $\lambda$), as per the discrete Stokes system (Section 2.1.1), the FE basis functions used to discretize the displacement $u$ and pressure $p$ cannot be chosen arbitrarily. Rather an inf-sup stable pair of FE basis functions must be used.

## 3 Incomplete $LDL^T$ (ILDL) preconditioning, with symmetric maximum weighted matching ordering, for saddle point matrices

Linear systems involving indefinite symmetric matrices are, in general, more difficult to solve than their positive-definite counterparts. This is partially due to the lack of positive-definite (inner product) structure. Diagonal entries which are small, zero, and/or not of constant sign make pivoting and numerical solution more challenging. Roughly speaking, though, if one can cast them as block systems which are in some sense better behaved, grouping problematic diagonal entries with better-behaved ones, robustness can be regained.

ILDL preconditioners arose as incomplete versions of *direct* solvers for indefinite systems, which use factorization $A = LDL^T$, after permutation and scaling. Here, $D$ is block diagonal with $1 \times 1$ and $2 \times 2$ blocks, and $L$ is lower triangular (Duff et al., 1991). Weighted matchings were first observed to be effective, *static* approximations to pivoting order by Olschowka and Neumaier (Olschowka and Neumaier, 1996). Duff and Koster introduced fast algorithms (Duff and Koster, 1999) and with the addition of Bunch-Kaufman pivoting (Bunch and Kaufman, 1977), highly efficient sparse direct solvers, both in terms of solution time and memory footprint, where made available (Duff and Pralet, 2005; Schenk and Gärtner, 2006). These have become the standard for the direct solution of sparse indefinite systems (Li and Demmel, 2003; Schenk and Gärtner, 2006).

By limiting the number of non-zeros (the "fill") in $L$, one can obtain an approximate factorization to be used as a precon-ditioner for a Krylov method (Hagemann and Schenk, 2006). Wubs and Thies present results for the special case of Stokes





$\mathcal{F}$-matrices, arising from a simple finite-difference scheme (Wubs and Thies, 2011). A closely-related approach which we do not investigate here is that of signed incomplete Cholesky factorization preconditioners (Scott and Tůma, 2014).

Permutation and scaling based on symmetric maximum weighted matching algorithms have shown to nearly or completely eliminate the need for pivoting in the factorization process, thus giving rise to very efficient methods (Duff and Pralet, 2005). We thus describe these considerations in further detail, to give a flavor of the sophisticated methods which are now available.

Numerical stability of incomplete factorization can be enhanced by permuting large elements onto the diagonal of a matrix. One may pose this task as a *(perfect) maximum weighted matching* procedure, finding a permutation ( a map from rows to columns) which maximizes the product of the absolute values of the diagonal entries. This can be accomplished via the Kuhn-Munkres algorithm (Laird and Giles, 2002; Munkres, 1957) with a complexity of $O(N^{1+\alpha} \log N), \alpha < 1$ for sparse matrices arising from finite difference or finite-element discretizations (Gupta and Ying, 1999); in practice, however the complexity

typically scales linearly with $N$ (Schenk and Gärtner, 2006).

If one wishes to find a *symmetric* permutation, one can only change the order of the diagonal entries. Nonetheless, one can extract cycles from the maximum matching and apply these symmetrically to move large entries *close* to the diagonal, in particular close to small or zero diagonal entries. If these cycles are decomposed into $1 \times 1$ and $2 \times 2$ cycles, one can then define a blocking wherein diagonal entries may be small, leading to poor conditioning, but $2 \times 2$ diagonal blocks have large

off-diagonal entries, making these blocks suitable pivots for a block elimination process (for more, and some useful diagrams, see (Bollhöfer et al., 2009, §2.2)).

This preprocessing is usually so effective as to not require any further pivoting (though additional Bunch-Kaufman pivoting is included in implementations, for maximum robustness) and in practice, the algorithm to solve the weighted-matching problem scales linearly in time, providing an extremely efficient method, far more attractive than methods without preprocessing steps.

Once this ordering preprocessing has been performed, a standard fill-reducing ordering may be performed on the full system. Thus, the complete factorization of a matrix $A$ may be represented as

$$\Pi^T \hat{P}^T \hat{D} A \hat{D} \hat{P} \Pi = A', \quad A' = LDL^T + E,$$

Where $\Pi$ is a fill-reducing permutation, $\hat{P}$ and $\hat{D}$ are a permutation and scaling arising from the symmetric maximum weighted matching preprocessing, and $E$ is an error introduced by the incomplete factorization process, which produces the incomplete factors $L$ and $D$ used in the preconditioner.

The ingredients in the preconditioner include the following components, each of which can be addressed separately in software.

– A reordering and scaling preprocessing step to reduce fill and the need for pivoting.

– An additional block-wise fill-reducing reordering

– A factorization stage which computes and stores $L$ and $D$, with respect to some drop tolerance, estimate of $||L||$, or specified fill pattern.

– A routine to quickly solve $LDL^T x = b$ by (block) forward- and back-substitution.




Despite the sophistication of the algorithms just discussed, practical usage of an ILDL preconditioner, given robust solver software, reduces to specification of only a few parameters; the user typically only needs to understand that there is a tradeoff between fill and the strength of the preconditioner, which they can control with some simple parameters (here, a drop tolerance).

## 4   Approximate block factorization (ABF) preconditioning

Approximate Block Factorization (ABF) solvers provide a powerful class of methods for the solution of saddle point systems.
These solvers define preconditioners by exploiting a block $LDU$ factorization of the saddle point matrix, with respect to the pressure and velocity blocks (Benzi et al., 2005, (5)). Approximately inverting the block-triangular or block-diagonal factors (often with available scalable solvers) provides a natural way to define approximate inverses, constructed from approximate solvers on a single field. For more, see details we refer to Elman et al. (2005).

    We choose a particular ABF solver as a representative of this class. In particular, we consider an upper block-triangular
preconditioner

$$\begin{bmatrix} K & B \\ & \hat{S} \end{bmatrix}, \tag{12}$$

where $\hat{S}$ is an approximation to the Schur complement $S$ given by $S = -C - B^T K^{-1} B$ (and noting that $C = 0$ in the Stokes case). The approximate solver on the viscous block is a geometric multigrid method, with a direct solve via UMFPACK (Davis, 2004) on the coarse level. Smoothing is accomplished by 8 Chebyshev-Jacobi iterations (Hu et al., 2003), where GMRES is
used to estimate the maximum eigenvalue $\lambda_{\max}$ of the preconditioned operator. The Chebyshev polynomial is tuned to the interval $[0.2\lambda_{\max}, 1.1\lambda_{\max}]$. The approximate Schur complement solver $\hat{S}$ is a single application of an ILU preconditioner formed from a scaled pressure mass matrix plus the $(2,2)$ block (which is zero in the case of Stokes); this is simple but is known to produce a spectrally-equivalent, hence scalable, preconditioner. This preconditioner was chosen based on experience using these solvers for applications in geodynamics, where it has shown to be scalable and efficient. For problems with large
non-grid-aligned coefficient jumps, more elaborate Schur complement preconditioners have also been developed in recent years (Elman, 1999; May and Moresi, 2008; Rudi et al., 2015, 2017). The ABF solver chosen here often shows superior performance for all but the smallest problem sizes, but relies on much more machinery set up in the application: the solver is aware of pressure and velocity blocks and a hierarchy of grids, transfer operators, and rediscretized operators. In addition, auxiliary operators must be defined to implement a Schur complement preconditioner, used here and in most competitive ABF solvers.

## 260  5   Numerical experiments

To provide a concrete and reproducible set of experiments, we use a $\mathbb{Q}_2 - \mathbb{Q}_1$ (Taylor-Hood) mixed finite element code[1], making use of the PETSC (Balay et al., 2019a, b) library. It solves the Stokes and elasticity systems in the unit square (2D) and

---

[1]Source code publicly available; see the "Code availability" section at the end of this paper, and the supplement on reproducibility.



cube (3D). We focus on 3D problems, as in the 2D case, sparse direct solution methods are expected to be highly competitive, with time to solution scaling as $O(N^{3/2})$ (as opposed to $O(N^2)$ for 3-dimensional problems) and expected fill scaling as

$O(N \log N)$ (as opposed to $O(N^2)$ for 3-dimensional problems) (George, 1973). For Stokes flow tests, free slip boundary conditions are imposed everywhere ($u \cdot n = 0, t \cdot \tau \cdot n = 0$) except the top boundary of the domain, where a free surface is prescribed ($n \cdot \sigma \cdot n = t \cdot \tau \cdot n = 0$); this implies a non-singular system matrix (Elman et al., 2005, Ch.5, p. 215). The elasticity tests also include experiments with non-zero Dirichlet boundary conditions (specified displacements).

Experiments involving both Stokes and elasticity systems are defined by a "multiple inclusion" configuration. That is, the

domain is partitioned into a set of $N$ non-overlapping spheres each with radius $R$. By providing parameters to control the (non-grid-aligned) coefficients (viscosity / Lamé parameters and density) contrast between the $N$ spherical inclusions and the surrounding medium, this model configuration provides a useful way to characterize two major factors which impact solver performance: coefficient jumps across arbitrary interfaces and the geometric complexity of these interfaces. Dealing with these factors is of primary important in designing solvers for realistic Earth science applications. This discontinuous coefficient field

is projected onto the quadrature points used to evaluate the bilinear / linear forms required by the finite element method. This projection has the effect of making the coefficient field vary on the length scale of a single finite elements; loosely speaking, this makes the problem "harder" and less amenable to solution with higher-order methods as the mesh is refined, but this is nonetheless consistent with the way that such problems are often solved in practice (Gerya and Yuen, 2003; May et al., 2015).

We compare three solver configurations: GMRES preconditioned with ILDL (see Section 3); sparse direct; and FGMRES

preconditioned with ABF (described in Section 4). We do not extensively compare to standard ILU preconditioning, or to ILDL preconditioning without symmetric maximum weighted matching preprocessing, as these preconditioners are very unreliable for indefinite problems (Chow and Saad, 1997). This characteristically poor performance has likely contributed to the fact that incomplete factorization preconditioners have not been championed, before this work, as a viable alternative for practical preconditioning, even though software tools have now developed to the point of making them robust options.

All linear algebra is dispatched through the PETSC API. We use a wrapper[2] to provide an interface between PETSC and internal functions in ILUPACK[3] which perform symmetric maximum weighted matching permutation and scaling, prior to a factorization step using a block elimination process with a simple threshold-based dropping strategy; entries in the $L$ factors less than a given value (after scaling and permutation) are dropped during the factorization process. Ordering with METIS (Karypis and Kumar, 1998) by nodes, with respect to the blocked system, proved robust and is used everywhere in this work.

In these experiments, available multi-level ILDL options did not seem to offer enhanced performance.

There are many packages for the sparse direct solution of linear systems; we choose PARDISO (Schenk and Gärtner, 2004; Schenk and Gärtner, 2006; Kuzmin et al., 2013) based on the comparative study of Gould et al. (2007) examining performance respect to total, serial (single-)solve time for symmetric indefinite systems with 10000 or more DOFs. Through a custom

---

[2]Source code publicly available; see the "Code availability" section at the end of this paper, and the supplement on reproducibility.

[3]Free academic licenses available; see the "Code availability" section at the end of this paper, and the supplement on reproducibility.



interface[4] we use PARDISO (Kuzmin et al., 2013) to provide a direct solver for symmetric indefinite systems, using the same
weighted-matching ordering used by ILUPACK and the ILDL preconditioners considered here.

All iterative solves use right-preconditioned GMRES or FGMRES and share a common convergence criterion: a reduction
of $10^6$ in the true residual 2-norm $\|b - Ax\|_2$. In practice, Krylov methods which take advantage of symmetric structure,
e.g. MINRES or QMR, may be attractive. The choice or norm allows is important because we consider ill-conditioned linear
operators for which convergence in a preconditioned norm often fails to imply convergence in the true residual norm. In
practice, different norms are usually used. These include preconditioned residual norms or a quasi-norm in the case of QMR
(Freund and Nachtigal, 1991).

Most of the computations were performed on single compute node of the Euler II cluster at ETH Zurich. Each compute node
is a dual-socket Intel Xeon E5-2680v3 nodes, each with 64 GB of memory. Numerical experiments used a single MPI-rank
and a single OpenMP thread. Experiments as reported in Figure 4 were performed on the Leonhard cluster at ETH Zurich,
using dual-socket Intel Xeon E5-2697v4 nodes, with 128 GB or more memory. Experiments as reported in Section 5.3 were
performed on Piz Daint at the Swiss National Supercomputing Center, using 6 MPI ranks per dual-socket Intel Xeon E5-2695v4
compute node with 64 GB of memory.

## 5.1   3D Stokes flow

Examples of 3-dimensional Stokes flow in cubic domains, for problem sizes ranging from $8^3$ to $64^3$ $\mathbb{Q}_2 - \mathbb{Q}_1$ elements,
are presented in Table 1 and Figure 1. Here, one can see comparable performance between the direct solve and the ILDL-
preconditioned solves, across all the problems tested; however, the ILDL-preconditioned solve requires less memory and has
the additional advantage of allowing for a loosening of the solve tolerance if desired. Due to its lower memory requirements,
we were able to solve larger problems with the ILDL-preconditioned approach. The ABF solver typically provides the best
time to solution, yet lacks robustness with respect to the problem parameters, in addition to relying on much more auxiliary
information and many more parameters (in particular, an auxiliary operator for the Schur complement preconditioner, and a
grid hierarchy and tuned parameters for the multigrid hierarchy).

As shown in Figure 1, increasing the number of inclusions degrades the performance of the ABF solver. This trend con-
tinues until eventually the ABF solver fails to converge. Figure 2 demonstrates the effect of further increasing the number of
inclusions, with a viscosity contrast of $\eta_1/\eta_0 = 10^6$ (a typical cutoff value for even-higher contrasts arising in geodynamical
modeling). Here we can directly observe a regime in which ILDL-preconditioned iterative solves not only provide a simpler
alternative to ABF solves, but a more robust one which also outperforms a direct solve in terms of time-to-solution and memory
footprint.

The iterative solver gives the user control over time-memory trade-offs by varying the drop tolerance. Table 2 shows the
effect of varying to drop tolerance with a $32^3$ element experiment as pictured in Figure 1(a). Comparable times to solution are

---

[4]Source code publicly available; see the "Code availability" section at the end of this paper, and the supplement on reproducibility.



(a) $\eta_1/\eta_0 = 10^4$.

(b) $\eta_1/\eta_0 = 100$.

(Vel. scaled $1/3\times$)

(c) $\eta_1/\eta_0 = 10^4$.

(Vel. scaled $1/3\times$)

(d) $\eta_1/\eta_0 = 10^4$.

**Figure 1.** Additional experiments, analogous to those in Table 1. Solver performance is assessed in terms of degrees of freedom solved to the prescribed tolerance ($10^{-6}$ relative error in the true residual norm) over the solution time, and by peak memory footprint.



| | GMRES(60)/ILDL(1e-3) | | | | PARDISO | | FGMRES(30)/ABF | | | |
| Els. | Fill | Its. | Time [s] | Mem. [MB] | Time [s] | Mem. [MB] | Lvls. | Its. | Time [s] | Mem. [MB] |
|---|---|---|---|---|---|---|---|---|---|---|
| $8^3$ | 2.0 | 14 | 2.27E+00 | 127 | 1.42E+00 | 163 | 2 | 22 | 2.46E+00 | 125 |
| $16^3$ | 2.9 | 45 | 5.27E+01 | 851 | 3.78E+01 | 1743 | 3 | 24 | 2.71E+01 | 1104 |
| $24^3$ | 3.7 | 112 | 4.77E+02 | 3076 | 2.98E+02 | 7289 | 3 | 19 | 7.74E+01 | 2330 |
| $32^3$ | 4.6 | 226 | 1.96E+03 | 8100 | 1.55E+03 | 21856 | 4 | 17 | 2.04E+02 | 9587 |
| $40^3$ | 4.5 | 420 | 6.41E+03 | 15513 | 6.67E+03 | 52376 | 4 | 17 | 3.10E+02 | 10426 |
| $48^3$ | 5.5 | 568 | 1.31E+04 | 30126 | - | - | 4 | 16 | 6.33E+02 | 17842 |
| $56^3$ | - | - | - | - | - | - | 4 | 15 | 7.01E+02 | 28219 |
| $64^3$ | - | - | - | - | - | - | 5 | 19 | 6.28E+03 | 40519 |

**Table 1.** 3D stationary Stokes flow, with 3 denser (relative density 1.2) spherical inclusions of viscosity $\eta_1 = 10^4$ in a surrounding medium of viscosity $\eta_0 = 1$. Iteration counts accompany data points in the graph. Missing data correspond to runs which failed due to insufficient available memory. See also Figure 1.

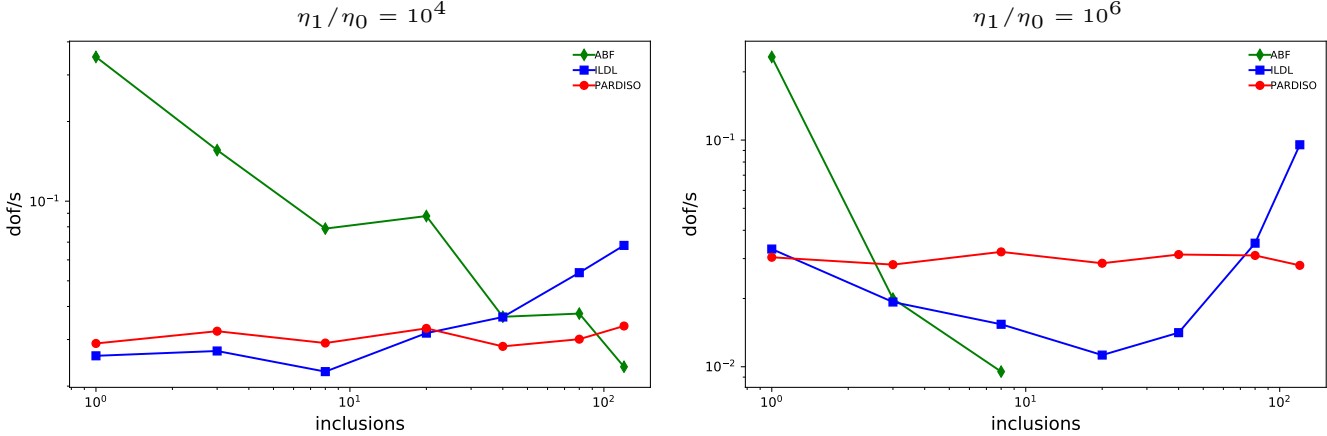

**Figure 2.** The effect of increasing the number of viscous inclusions on the effectiveness of solvers, for a $32^3$ element simulation. ILDL preconditioning becomes more competitive for more challenging structures, showing much greater robustness to viscosity structure than ABF solvers while maintaining a lower memory footprint than direct solution. Missing ABF data indicate that the solver failed to converge. Memory footprints are similar to those shown in Figure 1; the direct solve (red) requires approximately $4\times$ as much memory as the ILDL-preconditioned solve.





| drop tol. | fill | Solve Time [s] | Iterations | Max. Mem. [MB] |
|---|---|---|---|---|
| $1 \cdot 10^{-5}$ | 17 | 1.0696e+04 | 10 | 21072 |
| $5 \cdot 10^{-5}$ | 13 | 6.2200e+03 | 22 | 16582 |
| $1 \cdot 10^{-4}$ | 11 | 4.0494e+03 | 35 | 14321 |
| $5 \cdot 10^{-4}$ | 6.1 | 1.9075e+03 | 134 | 9666 |
| $1 \cdot 10^{-3}$ | 4.6 | 1.6031e+03 | 226 | 8433 |
| $5 \cdot 10^{-3}$ | 2.1 | 1.1421e+03 | 652 | 6585 |
| $1 \cdot 10^{-2}$ | 1.4 | 1.1657e+03 | 1015 | 6479 |
| $5 \cdot 10^{-2}$ | 0.39 | - | (>10000) | - |

**Table 2.** The effect of varying the drop tolerance parameter (see Section 3) for a $32^3$ element system and right-preconditioned GM-RES(60)/ILDL solver as in Table 1. Loosening the drop tolerance increases the iteration count and reduces the fill and hence memory footprint.

observed over a fairly broad range of drop tolerances, and experiments like this lead us to recommend default drop tolerances in the $10^{-3} - 10^{-4}$ range[5].

## 5.2   3D elasticity experiments

Figure 3 shows the results of similar experiments to those in Section 5.1 with spherical inclusions with different material properties, here varying $\lambda$ inside the inclusions, creating areas of near-incompressibility. This corresponds to a very high Poisson ratio $\nu = \frac{\lambda}{2(\lambda+\mu)}$ close to $\frac{1}{2}$, the challenging case where standard (non-mixed) finite element methods tend to exhibit locking. The effect of varying this parameter is markedly different to that of varying $\eta$ in the Stokes case (analogous to $\mu$ in the Lamé case); performance is not as dependent on this parameter for any of the solvers, and the attained dof/s for the ILDL solver degrades much more slowly. Again, we observe marked reduction in memory consumption for the ILDL solver, as compared to the direct solver, along with a notable performance gain from using ILDL preconditioning, relative to direct solution. ABF preconditioning is still advantageous, but may not be available in practice and may require an expert to set up. We note that the ABF preconditioner is not identical to that used for the Stokes problem, with $\mu$ substituted for $\eta$; to achieve good performance, the Schur complement preconditioner is different, due to the nonzero (2,2) block in the elasticity system. That is, to obtain a spectrally equivalent preconditioner, we needed to build a preconditioner $\hat{S}$ (Cf. equation 12) as an approximate inverse of $M_\mu - C$, where $M_\mu$ is a finite element mass matrix weighted by $\mu$ and $C$ is the term (depending on $\lambda$ as in Eq. (9)).

The experiments in Figure 3 are chosen to emphasize the similarity of the elasticity and Stokes problems, yet elasticity problems are commonly prescribed with non-zero boundary displacements, amounting to non-zero Dirichlet boundary conditions. Figure 4 shows a similar experiment using a scenario which is perhaps more typical in applications. The results parallel those in the previous set of experiments; the more-invasive ABF solver produces the fastest solution, but preconditioning with ILDL

---

[5]Note that dropping is performed on a system which has already been scaled, hence these values have meaning independent of the scaling of the system.

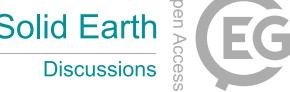

(a) 3 inclusions, $\lambda_1/\lambda_0 = 10^4$

(b) 1 inclusion, $\lambda_1/\lambda_0 = 10^2$

(c) 1 inclusion, $\lambda_1/\lambda_0 = 10^4$

(d) 8 inclusions, $\lambda_1/\lambda_0 = 10^4$

**Figure 3.** Experiments showing performance of ILDL preconditioning for the elasticity system, holding $\mu_0 = \mu_1 = 1, \lambda_0 = 1$ constant and varying $\lambda_1$ inside inclusions. Arrows show displacement, and the pressure field is plotted volumetrically.





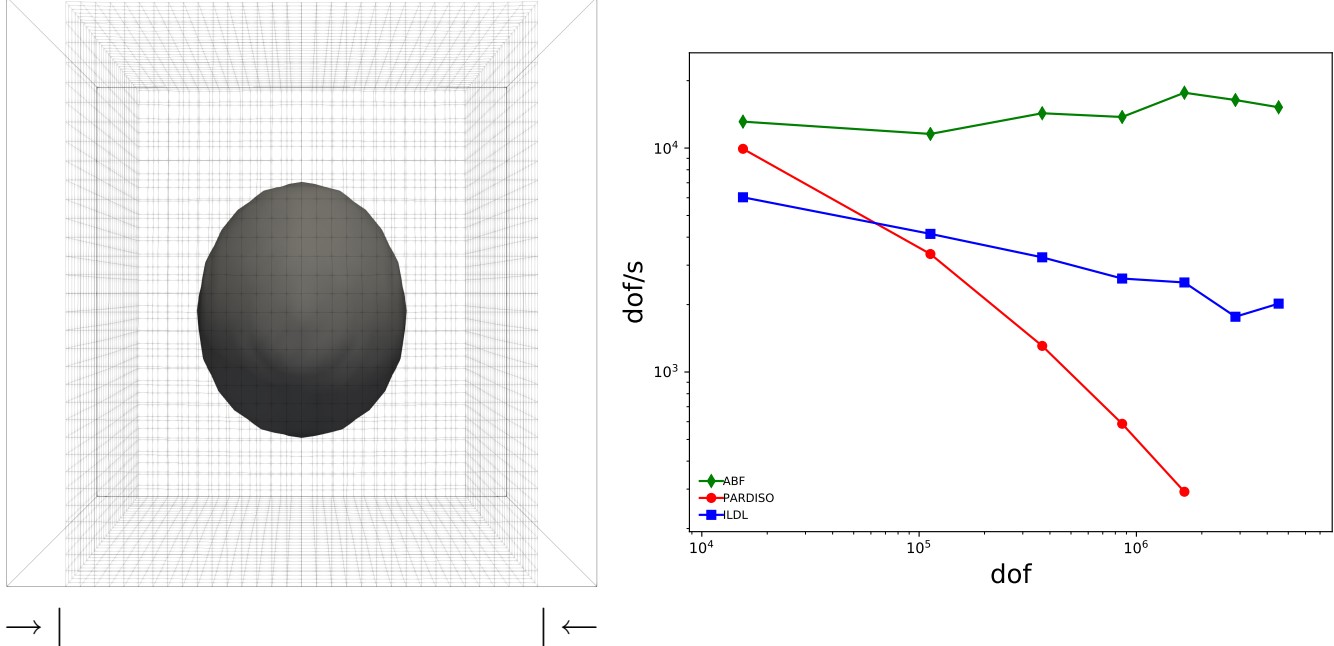

**Figure 4.** An experiment showing the performance of ILDL preconditioning for an elasticity problem with a heterogeneous medium in compression; the outer box shows the reference (undeformed) state and the wire mesh shows the deformed state. The surrounding medium has a Poisson ratio of $1/3$ ($\lambda = 2, \mu = 1$), and the originally-spherical inclusion is almost incompressible, with a Poisson ratio greater than $0.4999$ ($\lambda = 10^4, \mu = 1$). Boundary conditions are free-slip everywhere except the top, which is stress free. The results show that ILDL preconditioning offers substantially better-scaling performance (with a lower memory footprint) than a direct solver, without the auxiliary information, implementation, and tuning required for the even-better-scaling ABF solver. These experiments were run on a slightly different cluster than the preceding ones (see Section 5, so solve times are not directly comparable.

and symmetric weighted matching preprocessing gives a vastly superior option to a direct solve, while using no additional
information beyond the specification of a single drop tolerance. Memory usage is very similar to the plots shown in Figure 3.

### 5.3   Using ILDL within a parallel preconditioner

An obvious limitation of the results presented thus far, and of the particular implementation of the ILDL decomposition that we have employed, is that they have focused on single-process (i.e. "sequential") usage. However, most scientific computation is now performed with some degree of multi-process (or multi-thread) parallelism.

A well-known and often-used approach to extend a sequential preconditioner to a parallel preconditioner is to employ a domain decomposition method (Smith et al., 2004) wherein the computational domain is decomposed into possibly-overlapping patches where local preconditioners can be applied before the results are used to update the global solution. The simplest such preconditioner is the block Jacobi method, with non-overlapping subdomains, and a natural extension is the Additive Schwarz Method, wherein overlapping subdomains, here defined in terms of finite elements, are used to define subsolves.





ILDL preconditioning can be used to provide approximate inverses to local subproblems, much as ILU preconditioning is commonly used in the corresponding positive-definite case. We note that this block Jacobi/ILU preconditioning is practical useful for a wide range of problems despite, like ILU itself, it not being perfectly scalable or robust.

This is possible by leveraging the same software wrappers used in the sequential experiments in this paper, and indeed within the code used here, amounts simply to specifying a few command line options. As a proof of concept, table 3 compares iteration
counts and time-to-solution solving a Stokes problem with a block Jacobi or ASM preconditioner, with various choice of sub-preconditioner. As in the rest of this work, comparison is made with a well-tuned ABF solver. [6] No attempt has been made to optimize the subdomain solvers here - the drop tolerance was simply adopted from the sequential case. For the isoviscous case shown, the block Jacobi method and ASM method are comparable, but when a viscosity jump is added, the ASM solver can still converge, albeit slowly, while the block Jacobi approach becomes much slower. These simple experiments demonstrate that
ILDL preconditioners can be used within parallel preconditioners to solve problems too large for a single computational node. As in the sequential case, one can sacrifice some performance, with respect to a complex solver relying on more machinery and great expertise in tuning (ABF) to be able to quickly use a simpler (from the user's perspective) and more widely applicable solver.

## 6   Conclusions

The efficient solution of symmetric indefinite linear systems is an important task in many physical modelling applications in the Earth sciences and beyond, particularly solving PDE in mixed formulations. Approximate Block Factorization (ABF) preconditioned solvers (or other scalable options) including nested multilevel solves are well-known to be efficient for sufficiently-large problems, but require invasive code modifications and expertise in implementation and tuning, so might not be practical to implement or evaluate. Further, these solvers may not be robust to challenging coefficient structures. The ABF solver we
used for comparison here, for example, performs extremely well for small numbers of viscous inclusions but fails to converge for larger numbers. Advances in algorithms and software for direct solution of symmetric indefinite systems have, in recent years, brought direct solution for these systems to a level of performance and robustness on par with their counterparts for definite systems. These advances carry over to incomplete factorization preconditioning, though this is much less well-known; this work presents much-needed results on the effectiveness of these solvers in challenging parameter regimes. The Krylov
methods studied here, using ILDL preconditioning with a symmetric maximum weighted matching preprocessing step, require only a single drop tolerance parameter for the preconditioner, and can be useful across problem types, as seen here between Stokes and elasticity, and indeed even across "matrix market"-style corpora (Hagemann and Schenk, 2006; Greif et al., 2015). They also show useful robustness to viscosity structure, outperforming our representative ABF solver for larger numbers of viscous inclusions. Further, ILDL-preconditioned Krylov methods can be a preferable choice to direct solves: one trades some
parameter selection and less-robust performance for a large reduction in memory footprint and often extra performance.

---

[6]And note that the largest ABF solve here required the use of PCTELESCOPE (May et al., 2016) to define the coarsest grid on a subset of ranks, another wrinkle in the effective use of multigrid solvers.





| $\eta_1/\eta_0$ | Els. | MPI Ranks | GMRES(60)/Block Jacobi/ILDL(1e-3) | | GMRES(60)/ASM/ILDL(1e-3) | | FGMRES(30)/ABF | | |
|---|---|---|---|---|---|---|---|---|---|
| | | | Its. | Time [s] | Its. | Time [s] | Lvls. | Its. | Time [s] |
| | $32^3$ | 8 | 219 | 8.0564e+01 | 84 | 8.2991e+01 | 4 | 9 | 1.2685e+01 |
| | $48^3$ | 27 | 693 | 1.7149e+02 | 586 | 2.4185e+02 | 4 | 9 | 1.5954e+01 |
| | $64^3$ | 64 | 1310 | 2.6762e+02 | 1843 | 5.2592e+02 | 5 | 9 | 1.3151e+01 |
| | $96^3$ | 216 | 4668 | 7.6356e+02 | 7903 | 1.6136e+03 | 5 | 9 | 1.6056e+01 |
| 1 | | 125 | | | 4613 | 5.9810e+03 | | | |
| | $128^3$ | 216 | | | 6587 | 3.6521e+03 | | | |
| | | 512 | 6126 | 1.1899e+03 | 15396 | 4.8914e+03 | 5 | 9 | 1.3296e+01 |
| | | 729 | | | 19387 | 4.3029e+03 | | | |
| | $32^3$ | 8 | 709 | 1.6289e+02 | 155 | 9.3547e+01 | 4 | 12 | 2.1862e+01 |
| | $48^3$ | 27 | 14892 | 2.5563e+03 | 832 | 2.8261e+02 | 4 | 12 | 2.6607e+01 |
| $10^2$ | $64^3$ | 64 | - | $> 1.4e+04$ | 17636 | 6.9489e+03 | 5 | 14 | 2.1098e+01 |
| | $96^3$ | 216 | - | $> 2.8e+04$ | $> 10^5$ | 2.5548e+04 | 5 | 13 | 2.3065e+01 |
| | $128^3$ | 512 | - | $> 2.8e+04$ | $> 10^5$ | 2.6805e+04 | 5 | 14 | 2.3120e+01 |

**Table 3.** Data for MPI parallel solves, using a block Jacobi and 1-element overlapping ASM preconditioners, each with ILDL subdomain preconditioners, compared with an ABF solver as used throughout this paper. These solves correspond to the single-sinker Stokes case as in Figure 1(a)-(b), but now with larger problem sizes made possible by the distributed memory-environment. These show the feasibility of using incomplete factorizations to create an simple-to-apply parallel preconditioner for symmetric saddle point systems, albeit one which shows the same non-optimal scaling and parameter sensitivity familiar from the use of subdomain ILU or ICC preconditioners in parallel for definite systems. The problem sizes and number of ranks are chosen to demonstrate weak scaling (constant problem size per rank), and a strong scaling test is shown for one problem. Note that these experiments were conducted on a different cluster than those in previous sections (see Section 5), so times to solution are not directly comparable.

The results in this paper show that if one is employing a direct solver for a symmetric, indefinite problem, such as a Stokes or elasticity problem, an ILDL-preconditioned iterative solver is worth investigating. The preconditioner requires only an assembled operator and can be quickly used in situations that an ABF solver must be arduously selected, integrated, and tuned, and can offer greater robustness to coefficient structure. An ILDL-preconditioned iterative solver typically remains competitive

or even superior to direct solution, in terms of dof/s computed for larger problems while using $3 \times -5\times$ less memory. This alternative can be investigated quickly as only an assembled operator and one or two parameters need be provided.

We have focused on single-level ILDL preconditioning, but ILUPACK includes a multilevel ILDL preconditioner (Bollhöfer and Saad, 2006; Schenk et al., 2008), available through the same wrappers, and PARDISO (Schenk and Gärtner, 2004; Kuzmin et al., 2013) also includes a multi-recursive iterative solver which uses multi-level ILDL preconditioning. These techniques are

related to an algebraic multigrid (AMG) approach; further investigation is warranted of these and other highly-automated AMG approaches (Metsch, 2013) which may provide the scalability associated with multilevel methods while retaining the robustness and ease of use associated with factorization-based approaches. We have focused on sequential computation and application of

ILDL factorizations, and shown how these may be extended to the parallel case by using simple domain-decomposition-based preconditioners.

However, recent work has shown the promise of computing and applying incomplete $LDL^T$ factorizations directly in modern parallel environments, taking advantage of multiple threads, distributed memory subdomains (MPI ranks), and/or GPUs (Aliaga et al., 2014, 2016a, b, 2017; Bollhöfer et al., 2019b). As these developments make their way into software packages, these algorithms will become even more attractive for applications in the Earth sciences and beyond.

*Code availability.* Our solvers utilize functionality from the following libraries: ILUPACK (http://ilupack.tu-bs.de); PARDISO (www.
pardiso-project.org); PETSc (www.mcs.anl.gov/petsc). PETSc is open-source under a BSD-2 license; ILUPACK and PARDISO are closed source and offer complimentary academic licenses.

PETSc represents the highest level within our solver stack; all underlying solver implementations are utilized through PETSc function calls (e.g. `KSPSolve()`). To support this, we provide a wrapper around ILUPACK so that it can be used as a preconditioner (`PC`) implementation within PETSc (http://bitbucket.org/psanan/PCILUPACK). Through a custom interface we use PARDISO to provide a direct solver for
symmetric indefinite systems, using the same weighted-matching ordering )https://bitbucket.org/psanan/petsc/branch/psanan/pardiso-3.12.4). The code which performs the discretization of the Stokes and elasticity problems, configures the solvers and generates the post-processed flow/displacements fields is available here: http://bitbucket.org/psanan/exsaddle.

Also see the supplement to this paper, which provides additional instructions and details to reproduce, extend, and apply the experiments and tools discussed above.

*Author contributions.* All authors contributed collaboratively to the development of the project. P. Sanan, D. May and O. Schenk contributed to the writing of the manuscript. P. Sanan and D. May conceived and designed the scope and experiments presented, including the demonstration code. M. Bollhöfer provided code and support to interface with ILUPACK.

*Competing interests.* We declare there are no competing or conflicts of interest.

*Acknowledgements.* P. Sanan acknowledges financial support from the Swiss University Conference and the Swiss Council of Federal
Institutes of Technology through the Platform for Advanced Scientific Computing (PASC) program. D. May acknowledges financial support from the European Research Council under the European Union's Seventh Framework Programme (FP7/2007–2013)/ERC Grant Agreement Number 279925 and the Alfred P. Sloan Foundation through the Deep Carbon Observatory (DCO) Modeling and Visualization Forum. P. Sanan acknowledges Radim Janalik who helped developed the PETSc interface to PARDISO.



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
