# Peer review of "Pragmatic Solvers for 3D Stokes and Elasticity Problems with Heterogeneous Coefficients: Evaluating Modern Incomplete $LDL^{T}$ Preconditioners"

_Solid Earth, 2020_

## Referee Comment (RC1) · Marcin Dabrowski (Referee) · 25 Jun 2020

General comments

The paper addresses problems relevant to the scope of SE and includes some interesting novel concepts regarding the numerical solution of 3D mechanical problems. The scientific methods and assumptions are sound and the presented conclusions are justified. The authors give credit to previous related work and they clearly delineated their contribution. The paper is well written and properly structured, the title is informative

and clear, and the abstract provides a good summary of the work.

Below I present my specific comments and technical corrections. I would encourage the author to include a more detailed presentation of the studied numerical setups within the main body of the manuscript, according to my detailed suggestions below.

Specific comments

For the Taylor-Hood element, the static elasticity in the mixed finite element formulation produces a symmetric indefinite system. It is maybe worth noting that for FEM discretization with piecewise discontinuous pressure field such as in the case of the Crouzeix-Raviart element family, the pressure mass matrix can be easily inverted on the element level and by performing block Guassian elimination a positive definite system can be obtained that allows for using the highly robust sparse Cholesky factorization.

The author claimed that the previous incarnations of ILDL' were not necessarily robust (l.93-94). Could the authors just briefly mentioned the major improvements within the recent ILDL' implementation? What improvements exactly have made them robust in the recent years?

What is exactly meant by "coefficient structure" (for example l. 115)? I guess that this is not just the sparsity pattern.

1x1 and 2x2 blocks are mentioned in the context of pivoting for both LDL' and ILDL'. I am actually wondering whether the natural blocking inherent to the problem due to its dimensionality is retained during this operation? It is stated that fill-reducing reordering is performed block-wise. Which blocks are exactly meant here? I would guess that the ones related to the problem dimensional (say 3x3 blocks in the case of 3D problems) How is it ensure that the blocking due to the symmetric maximum weighted matching preprocessing is retained during the subsequent fill-in reducing reordering? I would suggest that this issue could be clarified in the manuscript.

So what is exactly used as the Schur complement preconditioner for large coefficient jumps? The author mention "a scaled pressure mass matrix" in this context. What (viscosity) scaling is exactly used? If there is not enough space for explaining it, maybe the authors could refer to some other work here.

The authors claim that sparse direct solution methods for indefinite systems using LDL' are expected to be highly competitive for 2D cases (l. 263). Is there any recent study showing their real performance (not just the theoretical scaling) that could be referred here?

The authors make statements that variable coefficients on the level of individual elements (non-grid aligned coefficient jumps) are, loosely speaking, harder to solve. In what sense? Solution accuracy or solution time, or maybe both?

I would guess that referring to incomplete factorization preconditioners in l. 283, the authors specifically mean ILDL' rather than ILU or ICHOL, and that they perhaps make this statement in the context of geodynamics or, in general, geosciences.

The comparative study of Gould (2007) is mentioned in l. 292 to justify the choice of PARDISO. I am wondering where there could be any more up-to-date performance studies for sparse direct solvers of symmetric indefinite systems.

I totally agree with the authors that the choice of norm is important for matrices characterized by large condition numbers such as in the case of the studied Stokes problem with strongly variable viscosities. In this respect, the authors choose to use the true residual 2-norm rather than the norm induced by the preconditioning. It is perhaps outside the scope of this study, but, in my view, given that the authors have access to highly accurate solutions obtained using the direct solver approach, it would be quite interesting to check and compare the solution error between GMRES(60)/ILDL & FGMRES(30)/ABF, say in the energy norm.

Do the solve times reported in the tables for the ILDL' preconditioning include the time

spent on computing the ILDL' preconditioner? Actually, it would interesting to see how this time compares to the time spent on iterations.

Regarding the numerical setup, I would claim that what really matters is the fraction of the inclusion. With increasing inclusion fraction, as in the case of the setup studied in Fig. 2, a natural transition towards porous media like systems occurs (technically speaking, I am wondering how well 100 inclusions can be resolved using a 32^3 computational mesh). Such physical systems are characterized by strongly localized flows, which might be harder to solve compared to the suspension type of flow typically obtained for low concentration. It would be actually interesting to see how well the presented methods work when the gravity load is replaced by an ambient pressure gradient prescribed through boundary traction.

Technical corrections

l. 11-14 This sentence seems a bit convoluted. I would actually guess that something might be missing here.

l. 22-23 . . . the coefficient structure is made increasingly challenging – I would suggest formulating it more precisely; What "complex topologies: have been addressed in this study?

l. 33 This is maybe not so critical, but compressible quasi-static linear elasticity is not exactly an example of a problem with a divergence free displacement field. In addition, it may indeed represent a saddle point problem, but in some numerical formulations it may be straightforwardly cast as a positive-definite problem.

l.71 . . . the nonzero entries of the factors are restricted to those for A^k – This could be stated a bit more precisely.

l.72-73 I find the end part of this sentence unclear.

l. 109 In contrast to the previous ILDL studies previous mentioned above . . . - please remove the second instance of "previous"

l. 134 One could consider using the transpose for one of the vectors in n*\sigma*n, etc in eq. 3 & 4.

l. 179 Is it really necessary to replace \tau with dev(\sigma) in eq. 3? Given that the t and n vectors are perpendicular (<t,n> = 0), t'*sigma*n=t'*(tau-p*I)*n = t'*tau*n-p*t'*n = t'*tau*n

l. 210 permutation ( a map from rows to columns ) – I would think that the permutation operates within the rows and within the columns, and not from rows to column.

l. 215 If one wishes to find a symmetric permutation, one can only change the order of the diagonal entries. – If I am getting it right, a symmetric permutation preserves the symmetry of the matrix. I guess that with changing the order of the diagonal entries, the order of the entire rows and columns is also changes (not just the order of the diagonal entries). Anyway, could "non-symmetric" permutations be considered in the current context?

l. 293-4 Through a custom interface we use PARDISO (Kuzmin et al., 2013) – This looks a bit repetitive with respect to the previous sentence.

l. 298 The choice or norm allows is . . . - Please fix.

Table 1 - I would suggest that the volume fraction of the inclusions could be given. The viscosity is shown without the unit, and this problem could be easily solved by showing the viscosity ratio. Is the relative density dimensionless? Is it defined as (\rho_incl-\rho_host)/\rho_host? Is it actually relevant given that the model is linear? I would suspect that changing the relative density should only result in a rescaled velocity, and it should, hopefully, produce no appreciable changes to the course of numerical iterations. Is "fill" defined as the ratio between the non-zero entries in the ILDL' factor with respect to the non-zero entries of the original matrix (the triangular part of it, including the diagonal)? Is it necessary to use the scientific notation when time is reported? Maybe giving the total dof count could be useful.

Figure 1 - I would suggest a more detailed description of the numerical setups, both in the caption and in the main body of the manuscript. What is the volume fraction of the inclusions? What is meant by (Vel. scaled 1/3x)? Isn't it that the scaling of the quiver lengths is in no obvious way absolute? I think that it would be useful to show grid lines in the plots. I guess that the dashed line in the Peak Memory Footprint shows the maximum available RAM during the numerical tests, but it would be useful to explain it in the caption. The curve styles are not well visible in the legend. It could also be explicitly explained in the caption that ABF(a), ABF(b), . . . refer to setups a, b, c. . . (at a first glance it may look as if it were some variants of the solvers).

l. 317-8 . . . the ABF solver fails to converge. – It is not clear to me where this can be seen in Fig. 1 (I can't really see any missing data for ABF)

Figure 2 - What is the volume fraction of the inclusions as their number is increase? Given that the numerical resolution is kept constant (32ˆ3) I would guess that it is increased. In my opinion, this should be explicitly stated in the caption and also in the main body of the manuscript. In fig. 1 for 32ˆ3 the overall solver performance in terms of dof/s fell in to the range between 5*10ˆ3 and 10ˆ4, which is consistent with the time reported in table 1. However, in fig. 2, even in the previously studied case of the viscosity ratio of 10ˆ4, the performance is between 10ˆ-2 and 10ˆ-1. I would guess that this could be some technical mistake. In my opinion, it would be useful to show gridlines and maybe use a slightly large font for the legend entries.

l.324 ". . . varying to drop tolerance" – Please fix.

l. 326 System scaling is mentioned in the footnote. Please explain what system scaling (physical, algebraic, ..) is exactly meant here.

l.339.. and C is the term (depending on \lambda as in Eq.(9)). – I would guess that the outer brackets are not necessary here. Could the author hint what they actually use for the C term?

l. 340 Figure 4 shows a similar experiment using a scenario which is perhaps more typical in applications. – Please explain the boundary conditions used in this setup in the main body of the manuscript.

Figure 3 –Maybe the Lame parameters \mu and \lambda could be scaled by \rho*g*L. A colorbar for the color-coded pressure and gridlines would be a nice addition to this figure.

Figure 4 – It is of small relevance to the studied topic, but the deformed wire mesh implies a substantial deformation that could hardly be accommodated elastically by any geomaterial. But maybe this could be treated as an exaggerated mesh deformation. The elastic moduli are given with no units.

Marcin Dabrowski

---

## Referee Comment (RC2) · Mikito Furuichi (Referee) · 14 Jul 2020

General comments:

This paper presents the benchmark experiment with the direct and iterative solvers for the Stokes flow and elastic problems targeted by the solid earth simulation. The authors especially focus on the ILDL factorization which is not yet commonly used in the numerical solid earth community. Their performance test showed the tradeoff relations among the robustness, time to solution, and memory cost. This paper is well

organized and presented results may motivate the computational geoscientist to utilize ILDL in their own geodynamics and seismic applications. Thus, this paper essentially fits the scope of the method paper of Solid Earth (SE).

On the other hand, there is some room for improvement in presentation and experimental design. The author claims that the robustness of ILDL is the advantage over the iterative ABF solver, but supporting experimental data is found only in the extreme case which solves 10 inclusions of $10^6$ viscosity contrast within $32^3$ elements simulation. In other cases, iterative ABF solver shows better results in time-to-solution, memory usage, parallel performance. On the other hand, ILDL shows practical advantages against direct solver in memory cost. Thus, in conclusion, the ILDL solver is found to be the potentially good alternative of direct solver rather than an iterative solver. So, the expected reader would be the user of direct solver. However, their performance analysis is presented mainly for ILDL vs iterative ABF solver rather than vs direct solver, especially in the parallel performance section. I encourage the author to continue this work, but the presentation should be improved and more detailed performance analysis should be addressed before I recommend this for publication in SE.

Detail comments:

1. In introduction: Several sentences sound your opinion rather than the objective view (e.g. "This is unfortunate" in line 102). Such phrases are not appropriate for the research paper.

2. In introduction: Please more review the progress and difficulty in direct solvers, although the author mainly reviews the recent progress of iterative solver.

3. In line 45: The hieratical grid system with such as AMR [Rudi et.al. 2015] worked well as the solution of highly variable viscosity problem with controlling the coefficient.

4. In line 95: Since expected readers of this journal are not specialists in linear alge-

bra, a more comprehensive review is needed. For example, how much memory was saved against direct solver with increasing/decreasing the time-to-solution in the past successful application?

5. In line 211: Delete the space after "( "

6. In line 253: The spectral analysis for scaled pressure mass matrix can cite [1]

7. In Numerical experiment: I think that the experiment starts with x=0. This problem setting is suitable for steady-state solution. But in practice, we solve the time-stepping/nonlinear problems. Thus, it is interesting if ILDL largely outperforms the direct solver from the second step. The solution of the previous step will be a good initial gauss for reducing the iteration of ABF and ILDL.

8. In line 264: In practice, direct solver is mainly used in 2D problems. Also, in memory capacity, the difference in maximum element size in 3D (40ˆ3 for PARADISO < 48ˆ3 for ILDL) seems to be trivial but that in 2D (252ˆ2 < 332ˆ2) is significant in scientific application. Then, the experiment in 2D should worth considering in SE.

9. In line 317: It is confusing that ABF does not fail to converge in Figure1. Why not plot the case with contrast = 10ˆ6 with 8 inclusions?

10. In line 318: Do we really need to solve the problem with over 10 inclusions in 32ˆ3? The accuracy of such a setting seems to be a useless solution in physics. In addition, to check the robustness, SINKER box test of [May and Moresi, 2008] is better than this setting.

11. In Figure 1: Sample glyphs are difficult to see.

12. In Figure 1: What is the message from the peak memory foot point? Why memory size in Table2 is not enough?

13. In Table2: For a fair performance comparison, it should be noted that the number of iterations independent from the DOF for ABF.

14. In line 320: Since your ABF is based on Jacobi smoother and Arnoldi type Krylov method, more smoothing iteration or avoiding rounding error of GMRES are promising to gain the convergence even with 10ˆ6 problem. It is interesting to see the performance of ABF with increasing the number of inner smoothing iterations to converge 10ˆ6 problem (I argue that such simple tuning is out of the expertise.). Whether such robust ABF can solve the 10ˆ6 problem faster than the ILDL method or not, is the matter for ILDL to be the alternative of ABF.

15. In line 297: Please write Eqs. (5), (11), and the norm should be a consistent form.

16. In line 353: Additive Schwarz Method (ASM) should be noted.

17. In "Using ILDL within a parallel preconditioner": Since ILDL is worth investigating as an alternative of direct solver PRADISO rather than ABF solver, the performance on SMP system (openMP) is more interesting than distributed memory parallelization (MPI). Please reconsider the way of presentation. Since ABF is inherently suitable for the distributed memory parallelization, Table3 did not show any advantage of ILDL.

18. In lines in 400-404: These lines seem to be a jump in the context. Please introduce them in more detail if you want to address them. By the way, "incomplete LDL" should be ILDL

[1] P. P. Grinevich and M. A. Olshanskii, An iterative method for the Stokes-type problem with variable viscosity, SIAM Journal on Scientific Computing, 31 (2009), pp. 3959–3978

———————————————————

---

## Author Comment (AC1) · 22 Aug 2020

**se-20200-79-RC1**
**Response to Reviewer Comments**

Patrick Sanan, for all authors

We deeply thank the reviewer, Marcin Dabrowski, for insightful and thorough comments. We reproduce these comments here and provide our own comments and responses inline. Line numbers in our responses refer to the revised manuscript.

> General comments
>
> The paper addresses problems relevant to the scope of SE and includes some interesting novel concepts regarding the numerical solution of 3D mechanical problems. The scientific methods and assumptions are sound and the presented conclusions are justified. The authors give credit to previous related work and they clearly delineated their contribution. The paper is well written and properly structured, the title is informative and clear, and the abstract provides a good summary of the work. Below I present my specific comments and technical corrections. I would encourage the author to include a more detailed presentation of the studied numerical setups within the main body of the manuscript, according to my detailed suggestions below.

> Specific comments
> For the Taylor-Hood element, the static elasticity in the mixed finite element formulation produces a symmetric indefinite system. It is maybe worth noting that for FEM discretization with piecewise discontinuous pressure field such as in the case of the Crouzeix-Raviart element family, the pressure mass matrix can be easily inverted on the element level and by performing block Guassian elimination a positive definite system can be obtained that allows for using the highly robust sparse Cholesky factorization.

In the discussion of the ineffectiveness of ILU preconditioning (line 289), we have added a citation to Dabrowski et al., 2008 to highlight this approach, which indeed needed to be mentioned.

> The author claimed that the previous incarnations of ILDL' were not necessarily robust(l.93-94). Could the authors just briefly mentioned the major improvements within the recent ILDL' implementation? What improvements exactly have made them robust in the recent years?

This mainly refers to ILDL preconditioning without the weighted-matching step, or just applying ILU or incomplete Cholesky factorizations directly. As in the Chow and Saad 1997 reference cited on line 284, ILU factorizations perform very unreliably. Hagemann and Schenk's 2006 paper presents some experiments which compare the effect of different pivoting strategies.

> What is exactly meant by "coefficient structure" (for example l. 115)? I guess that this is not just the sparsity pattern.

This is intended to refer to the functional form of the coefficients, that is how the material properties vary over the domain. Of interest in the context of solver robustness is whether these coefficients have large global

variation, large local variations, and geometrically simple (roughly, "low frequency") or complex distribution of these variations.

We've further clarified the use of this term on line 113.

> 1x1 and 2x2 blocks are mentioned in the context of pivoting for both LDL' and ILDL'. I am actually wondering whether the natural blocking inherent to the problem due to its dimensionality is retained during this operation? It is stated that fill-reducing reordering is performed block-wise. Which blocks are exactly meant here? I would guess that the ones related to the problem dimensional (say 3x3 blocks in the case of 3D problems). How is it ensure that the blocking due to the symmetric maximum weighted matching preprocessing is retained during the subsequent fill-in reducing reordering? I would suggest that this issue could be clarified in the manuscript.

The natural blocking $(2 + 1$ or $3 + 1)$ is not directly retained during the factorization. All blocking and ordering at the level of the preconditioner is done with $2 \times 2$ and $1 \times 1$ blocks.

This has the advantage of requiring less information from the user, which is extremely desirable in the context of providing widely-applicable and robust methods. However, methods which take the specific saddle point structure of the problem into account do exist; for instance, the paper from Wubs and Thies (in the paper's references) takes into account what they call $\mathcal{F}$-matrix structure.

We have added a footnote (line 233) to emphasize which blocking is being discussed.

> So what is exactly used as the Schur complement preconditioner for large coefficient jumps? The author mention "a scaled pressure mass matrix" in this context. What (viscosity) scaling is exactly used? If there is not enough space for explaining it, maybe the authors could refer to some other work here.

This has been clarified in the text (line 345) to note the exact scaling, $-\left(\frac{1}{\mu} + \frac{1}{\lambda}\right)$. As suggested by Reviewer 2, we have added a reference (line 254) to Grinevich and Olshanskii's 2009 paper, which discusses this preconditioner. Also see the response below to the comments about the $C$ term.

> The authors claim that sparse direct solution methods for indefinite systems using LDL' are expected to be highly competitive for 2D cases (l. 263). Is there any recent study showing their real performance (not just the theoretical scaling) that could be referred here?

We include a 2D study, from an earlier draft of this paper, which hopefully gives a direct example of this, in Figure 1 of this response.

> The authors make statements that variable coefficients on the level of individual elements (non-grid aligned coefficient jumps) are, loosely speaking, harder to solve. In what sense? Solution accuracy or solution time, or maybe both?

With sharply-varying fields, higher order convergence is harder to obtain. In practice, solution times are likely longer, both due to this lower convergence order requiring a finer mesh, or by the fact that multigrid methods appear to converge less quickly.

> I would guess that referring to incomplete factorization preconditioners in l. 283, the authors specifically mean ILDL' rather than ILU or ICHOL, and that they perhaps make this statement in the context of geodynamics or, in general, geosciences.

Yes - this sentence was only meant to refer to incomplete factorization preconditioners for indefinite problems in geosciences (or even more generally in computational mechanics). We have updated it in the discussion around line 284.

[Figure]

| | GMRES(60)/ILDL(1e-3) | | | | PARDISO | | FGMRES(30)/ABF | | | |
|---|---|---|---|---|---|---|---|---|---|---|
| Els. | Fill | Its. | Time [s] | Mem. [MB] | Time [s] | Mem. [MB] | Lvls. | Its. | Time [s] | Mem. [MB] |
| $32^2$ | 2.9 | 36 | 4.34E-01 | 27 | 1.89E-01 | <1024 | 3 | 19 | 6.43E-01 | <1024 |
| $128^2$ | 4.4 | 229 | 2.37E+01 | 324 | 3.27E+00 | 404 | 5 | 15 | 5.68E+00 | 335.00 |
| $512^2$ | 5.7 | 1480 | 2.29E+03 | 5879 | 8.37E+01 | 7059 | 7 | 17 | 1.04E+02 | 5292.00 |
| $1024^2$ | | | | | 6.28E+02 | 31266 | 8 | 18 | 3.90E+02 | 20846.00 |

Figure 1: Experiments comparing solvers for a 2D stationary Stokes flow problem with dense $(1.1\times)$, viscous $(\eta_1 = 10^4, \eta_0 = 1)$ inclusions, discretized with $\mathbb{Q}_2 - \mathbb{Q}_1$ finite elements. Extremely short runs do not always report accurate memory usage on the test system.

> The comparative study of Gould (2007) is mentioned in line 292 to justify the choice of PARDISO. I am wondering where there could be any more up-to-date performance studies for sparse direct solvers of symmetric indefinite systems.

Unfortunately, we couldn't find a good, more-recent study (though this would also be of great interest to us). We hope that the older study at least convinces the reader that we have chosen a highly-competitive direct solver, to provide a meaningful comparison with other options. We have slightly modified the statement in the paper (line 300), to imply that we choose it as a competitive solver in general, and that the cited paper offers some concrete, though not complete, evidence of this.

> I totally agree with the authors that the choice of norm is important for matrices characterized by large condition numbers such as in the case of the studied Stokes problem with strongly variable viscosities. In this respect, the authors choose to use the true residual 2-norm rather than the norm induced by the preconditioning. It is perhaps outside the scope of this study, but, in my view, given that the authors have access to highly accurate solutions obtained using the direct solver approach, it would be quite interesting to check and compare the solution error between GMRES(60)/ILDL & FGMRES(30)/ABF, say in the energy norm.

Choosing a norm is usually a compromise, and here we chose one that we thought would be the most representative of something like "actual solver performance", in terms of minimizing a quantity (true residual norm) which most people would agree corresponds well to minimization of the quantity of actual interest. However, as noted in the paper (line 308), in practice a different (quasi-)norm would likely be used, either because of computational expedience or because it better represented the application's idea of an accurate solution. We do think it's probably outside the scope of the paper to present the results in additional norms, but agree that it would be interesting to interrogate and should be, if a practitioner relies on a specific norm.

As a related point here, we have devoted considerable effort to making the experiments here reproducible, with publicly-available (and usable under a BSD-2 license) source code, to help address the problem that users will often want or need to further interrogate the experiments presented. Here, additional norms could be examined by running the application code (with the help of our reproduction supplement), and using PETSC command line options to monitor a different norm or (if the desired norm is not supported), even modifying the C code. While obviously the investment of time and effort to run any code is non-trivial, as readers we very much appreciate the ability to examine source code to answer the common question of how a description in a paper ultimately translates to the implementation, and if interested in extending the method, to be able to directly compare to an existing implementation. As scientific software becomes more and more complex and relies on larger and larger software stacks, the notion that reproducibility (and re-implementability) is implied by a technically-complete description of algorithms becomes less and less valid.

> Do the solve times reported in the tables for the ILDL' preconditioning include the time spent on computing the ILDL' preconditioner? Actually, it would interesting to see how this time compares to the time spent on iterations.

The solve times include both the setup and solve time. The motivation for this is as mentioned on on line 114 - for most of the applications we envision being relevant (3-dimensional nonlinear problems with large-enough memory footprints that direct solvers become problematic), the system is only solved once.

The setup time usually dominates the application time, very roughly requiring $50 - 90\%$ of the total solve time, for the experiments in this paper. If running the included code, one can observe this by using PETSC's logging feature (use `-log_view` as an option), and then can observe the amount of time spent in `PCSetUp`, where the factorization is computed.

We agree that this is under-reported in the paper (especially given the prevalence of reporting these times separately in computational science literature, whether or not is really relevant). Thus, we have added a

note on line 297 and a new column of setup times in Table 1.

> Regarding the numerical setup, I would claim that what really matters is the fraction of the inclusion. With increasing inclusion fraction, as in the case of the setup studied in Fig. 2, a natural transition towards porous media like systems occurs (technically speaking, I am wondering how well 100 inclusions can be resolved using a 32ˆ3 computational mesh). Such physical systems are characterized by strongly localized flows, which might be harder to solve compared to the suspension type of flow typically obtained for low concentration. It would be actually interesting to see how well the presented methods work when the gravity load is replaced by an ambient pressure gradient prescribed through boundary traction.

None of the experiments presented here includes a particularly high volume fraction of inclusions.

The intent of these benchmarks is not to stress the solvers by imposing a complex domain (as for instance would be the case for a high volume fraction of rigid inclusions, with only the interstitial space meshed). Rather, it is to explore the solver performance on a simple domain, but with coefficient structures which vary across non-grid-aligned discontinuities. The multiple sinker benchmark is convenient, but one could also have explored, say, a sinusoidal boundary between two materials, varying the coefficient jump across the boundary, and the frequency of the boundary.

> Technical corrections
>
> l. 11-14 This sentence seems a bit convoluted. I would actually guess that something might be missing here.

A typo (extra "to") has been fixed and this sentence (line 11) has been simplified and split into two.

> l. 22-23...the coefficient structure is made increasingly challenging – I would suggest formulating it more precisely; What "complex topologies" have been addressed in this study?

We have changed this sentence to specifically mention the multiple-inclusion scenario (line 24).

> l. 33 This is maybe not so critical, but compressible quasi-static linear elasticity is not exactly an example of a problem with a divergence free displacement field. In addition, it may indeed represent a saddle point problem, but in some numerical formulations it may be straightforwardly cast as a positive-definite problem.

This was indeed incorrectly expressed - to amend this, we have added a sentence to mention the interesting and computationally-relevant fact that systems requiring divergence-free flow/displacement fields can be seen as limiting cases of systems which penalize volume changes (line 30).

We have chosen to focus on incompressible or nearly-so examples in this work, as outside of this context, there is less motivation to introduce elasticity in mixed form (even at the continuous level).

> l.71...the nonzero entries of the factors are restricted to those for Aˆk – This could be stated a bit more precisely.

Fixed to "$A^{k+1}$" and wording clarified (line 71).

> l.72-73 I find the end part of this sentence unclear.

The has been reworded to be more concise (line 72), as the point of this passage is simply to point out that only a small number of parameters are required for various variants of ILU preconditioning.

> l. 109 In contrast to the previous ILDL studies previous mentioned above...- please remove the second instance of "previous"

Fixed.

> l. 134 One could consider using the transpose for one of the vectors in $n * \sigma * n$, etc in eq. 3 & 4.

We opt not to do this, to try to keep the notation uncluttered. We believe it is at least consistent notation, in the sense that $u \cdot v = u^T v$ for two vectors, $u \cdot A = A^T u$ for a vector and a tensor (thought of as a matrix) and similarly $T \cdot u = Tu$.

To make this section more clear, we have added a description of the boundary conditions as a partition of the boundary into free-slip and free surface (zero stress) regions (line 134).

> l. 179 Is it really necessary to replace $\tau$ with $\mathrm{dev}(\sigma)$ in eq. 3? Given that the $t$ and $n$ vectors are perpendicular ($\langle t, n \rangle = 0$), $t' * \sigma * n = t' * (\tau - p * I) * n = t' * \tau * n - p * tn = t' * \tau * n$

This is true, and indeed this requires no special treatment in our code, so we have removed this statement.

> l. 210 permutation ( a map from rows to columns ) – I would think that the permutation operates within the rows and within the columns, and not from rows to column.

This was confusingly written and we've removed the mention of the row-column map (line 209), as the interested reader is better served by reading about the details of the matching in the references. Briefly, the problem of permuting the matrix is re-cast as a matching problem: the matrix is interpreted as a weighted bipartite graph, where entries correspond to edges between rows and columns. The maximum weighted matching gives (for a nonsingular matrix) a subset of the edges such that each row and column is involved exactly once, which can thus be interpreted as a permutation. This subset is sought which maximizes an objective (the product of the entries).

> l. 215 If one wishes to find a symmetric permutation, one can only change the order oft he diagonal entries. – If I am getting it right, a symmetric permutation preserves the symmetry of the matrix. I guess that with changing the order of the diagonal entries,the order of the entire rows and columns is also changes (not just the order of the diagonal entries). Anyway, could "non-symmetric" permutations be considered in the current context?

A symmetric permutation of a square matrix $M$ is indeed of the form $PMP^T$, where $P$ is a permutation matrix, and preserves the symmetry of the matrix while moving entire rows and columns.

We have not considered non-symmetric permutations in this context, but we do not categorically disregard them. The aim of the permutations is to provide a good pivoting strategy, and the current approach seems to scale close to optimally in practice while still retaining system symmetry (thus allowing one to store only $L$ and $D$, as opposed to two triangular factors, and allowing the use of methods like QMR or MINRES which require symmetric systems). Thus, exploring non-symmetric permutations becomes less appealing.

However, the question of relaxing the symmetry requirements is a very interesting one! On the practical level, this would be highly desirable for applications, for instance finite difference schemes (including finite volume schemes on orthogonal grids) for the Stokes equations, which don't produce a symmetric system.

Interesting future work could address the usage algorithms which can extend the approaches presented here (for instance, exploring the use of the multi-level ILU as implemented in ILUPACK) to non-symmetric, indefinite systems which arise in computational geosciences.

> l. 293-4 Through a custom interface we use PARDISO (Kuzmin et al., 2013) – This looks a bit repetitive with respect to the previous sentence.

Modified to be more concise (line 299).

> l. 298 The choice or norm allows is...- Please fix.

Fixed.

> Table 1 - I would suggest that the volume fraction of the inclusions could be given. The viscosity is shown without the unit, and this problem could be easily solved by showing the viscosity ratio. Is the relative density dimensionless? Is it defined as $(\rho_{\text{incl}} - \rho_{\text{host}})/\rho_{\text{host}}$? Is it actually relevant given that the model is linear? I would suspect that changing the relative density should only result in a rescaled velocity, and it should, hopefully, produce no appreciable changes to the course of numerical iterations. Is "fill" defined as the ratio between the non-zero entries in the ILDL' factor with respect to the non-zero entries of the original matrix (the triangular part of it, including the diagonal)? Is it necessary to use the scientific notation when time is reported? Maybe giving the total dof count could be useful.

We haven't focused on volume ratio (though several are computed in the remainder of this response), because these ratios are low and we don't believe that this is a factor stressing the solver. We choose the multiple-inclusion problem as a benchmark not because of its direct relevance in application (where higher volume fractions are a key consideration) but because of its usefulness as an abstraction of difficult coefficient structure. See also our response to Reviewer 2's general comments.

However, the presentation wasn't clear enough to make the volume fraction obvious to the reader. As such, we've added the inclusion radius to the captions of Figures 1,3, and 4. This information was also available at the very end of our reproduction supplement, where we have updated the command-line options to include the default (0.1) inclusion radius.

We have changed the caption of Table 1 to mention only the viscosity ratio, and added a column for total DOFs. The relative density is dimensionless, and is defined as $\rho_{\text{incl}}/\rho_{\text{host}}$. Indeed, as this is a linear problem, most simple scalings of any of the parameters have little effect on the solver performance, which motivates the fact that we do not focus heavily on units or scalings in this paper, but on variations in the material coefficients.

"Fill" is defined as the number of nonzeros in the $L$ factor in the $LDL^T$ factorization, relative to the number of nonzeros in the strictly upper-triangular part of the matrix being factored. We have added a note in the paper to make this concrete (line 201). Fill is not reported with scientific notation, though drop tolerance is (which we thought was more readable).

> Figure 1 - I would suggest a more detailed description of the numerical setups, both in the caption and in the main body of the manuscript. What is the volume fraction of the inclusions? What is meant by (Vel. scaled 1/3x)? Isn't it that the scaling of the quiver lengths is in no obvious way absolute? I think that it would be useful to show gridlines in the plots. I guess that the dashed line in the Peak Memory Footprint shows the maximum available RAM during the numerical tests, but it would be useful to explain it in the caption. The curve styles are not well visible in the legend. It could also be explicitly explained in the caption that ABF(a), ABF(b),...refer to setups a, b, c...(at a first glance it may look as if it were some variants of the solvers).

The volume fractions of the inclusions can be computed from the inclusion count and radius (which are specified in Section 1.7 in the reproduction supplement).

- Single sinker of radius 0.25 $\implies$ volume fraction of 0.06

- 3 sinkers of radius 0.1 $\implies$ volume fraction of 0.01

- 8 sinkers of radius 0.1 $\implies$ volume fraction of 0.03

These are of course low volume fractions, in the context of problems concerned with interstitial flow (where indeed, other models than a pure Stokes model may be appropriate, e.g. including a Darcy-type term). Our use of the multiple-inclusion problem is motivated by its usefulness as an abstraction of coefficient structure which can affect solver performance, as further discussed in our response to the general comments from the second reviewer.

The captions about the velocity scaling are meant to signify that the first two plots have the same scaling, and the second two a different one, but as pointed out, the absolute scaling of these velocities is not very meaningful, so we have removed these notes to reduce clutter.

Grid lines have been added to all graphs in the paper. The flow plots already have grid lines, though one has to zoom in to see them.

Notes have added to the plots specifying that the dashed black lines are indeed the maximum RAM available.

All legend entries have been modified to hopefully make the line styles more clear. The caption of Figure 1 has been changed to refer directly to the coefficient structures (a)-(d).

l. 317-8...the ABF solver fails to converge. – It is not clear to me where this can be seen in Fig. 1 (I can't really see any missing data for ABF)

This was intended to mean that when using even more inclusions than are presented in Figure 1 are added, the performance continues to degrade, and thus there are missing ABF entries in Figure 2 (right). This has been clarified (line 326).

Figure 2 - What is the volume fraction of the inclusions as their number is increase? Given that the numerical resolution is kept constant ($32^3$) I would guess that it is increased. In my opinion, this should be explicitly stated in the caption and also in the main body of the manuscript. In fig. 1 for $32^3$ the overall solver performance in terms of dof/s fell in to the range between $5*10^3$ and $10^4$, which is consistent with the time reported in table 1. However, in fig. 2, even in the previously studied case of the viscosity ratio of $10^4$, the performance is between $10^{-2}$ and $10^{-1}$. I would guess that this could be some technical mistake. In my opinion, it would be useful to show gridlines and maybe use a slightly large font for the legend entries.

The volume fraction indeed increases with the number of inclusions. The inclusion are of radius 0.05 (in the unit cube), and as such for the maximum of 140 inclusions, representing about 7% of the volume. We've added a note to this effect in the caption of Figure 2.

There was indeed an error in our plotting script (an error in the calculation for the total number of DOFs). We have fixed the error and made the y axes uniform between the two sub-plots in Figure 2.

The legends have been increased in size in Figure 2, and grid lines have been added to all graphs in the paper.

l.324 "...varying to drop tolerance" – Please fix.

Fixed.

l. 326 System scaling is mentioned in the footnote. Please explain what system scaling(physical, algebraic, ..) is exactly meant here.

This has been clarified (line 333) to refer to a (newly-numbered) equation in Section 3, describing the preprocessing performed before the drop tolerance is applied.

> l.339.. and C is the term (depending on $\lambda$ as in Eq.(9)). – I would guess that the outer brackets are not necessary here. Could the author hint what they actually use for the C term?

The parentheses were indeed a typo, and we agree that the presentation was unclear.

The matrix $-C$ arises from a term in the weak form like $-\int_\Omega q^T \frac{1}{\lambda} p\, dv$, hence is just another scaled pressure mass matrix, scaled with coefficient $-\frac{1}{\lambda}$, which is ultimately added to the mass matrix scaled with $-\frac{1}{\mu}$, that arises in the same way that the $-\frac{1}{\eta}$ term does for the Stokes problem (as an easy to compute yet spectrally equivalent approximation for the $-B^T K^{-1} B$ term in the Schur complement).

This passage has been reworded in the manuscript (line 343), and more explicit descriptions of the weightings for pressure mass matrices have been added elsewhere (e.g. line 253).

Readers who may be interested in reimplementing the method, or simply wanting to see the direct expression of the formulae in C code, can also examine the source code (at the time of this writing, see `femixedspace.c:2987` at `bitbucket.org/psanan/exsaddle`).

> l. 340 Figure 4 shows a similar experiment using a scenario which is perhaps more typical in applications. – Please explain the boundary conditions used in this setup in the main body of the manuscript.

We have added this description and a short note in the initial description of the elasticity problem to highlight that in this case we use inhomogeneous boundary conditions; these are a simple modification of the free-slip conditions, adding terms to the righthand side to specify a given normal displacement as opposed to a zero normal displacement.

> Figure 3 – Maybe the Lame parameters $\mu$ and $\lambda$ could be scaled by $\rho * g * L$. A colorbar for the color-coded pressure and gridlines would be a nice addition to this figure.

We have not focused on scaling parameters, as global scalings of these do not affect linear solves, and only relative scalings affects the solvers considered here; indeed, it is one of the great advantages of the ILDL preconditioners, shared with the direct solvers, that they can largely automatically address scaling issues. In each case, the maximum pressure (red) is about 1, and the minimum is a small number (corresponding to a zero pressure at the top, free surface). We have added color bars for the pressure fields, and grid lines for the graphs. There are already grid lines (albeit faint) in the 3d plots.

> Figure 4 – It is of small relevance to the studied topic, but the deformed wire mesh implies a substantial deformation that could hardly be accommodated elastically by any geomaterial. But maybe this could be treated as an exaggerated mesh deformation. The elastic moduli are given with no units.

This being a linear problem, it is indeed hopefully still a relevant (and easier to visualize) experiment, even when using an unrealistically-large deformation.

We have not emphasized the units or absolute values of the parameters, as while these are obviously of great importance in actual applications, the applicability and effectiveness of the solvers discussed in this paper are crucially dependent on relative coefficient variability, and essentially invariant to scalings.

> Marcin Dabrowski

---

## Author Comment (AC2) · 22 Aug 2020

**se-20200-79-RC2**
**Response to Reviewer Comments**

**Patrick Sanan, for all authors**

We thank the reviewer, Mikito Furuichi, very much for detailed and thoughtful comments. We reproduce these comments here and provide our own comments and responses. Line numbers in our responses refer to the revised manuscript.

> General comments:
>
> This paper presents the benchmark experiment with the direct and iterative solvers for the Stokes flow and elastic problems targeted by the solid earth simulation. The authors especially focus on the ILDL factorization which is not yet commonly used in the numerical solid earth community. Their performance test showed the tradeoff relations among the robustness, time to solution, and memory cost. This paper is well organized and presented results may motivate the computational geoscientist to utilize ILDL in their own geodynamics and seismic applications. Thus, this paper essentially fits the scope of the method paper of Solid Earth (SE). On the other hand, there is some room for improvement in presentation and experimental design. The author claims that the robustness of ILDL is the advantage over the iterative ABF solver, but supporting experimental data is found only in the extreme case which solves 10 inclusions of 10^6 viscosity contrast within 32^3 elements simulation. In other cases, iterative ABF solver shows better results in time-to-solution, memory usage, parallel performance. On the other hand, ILDL shows practical advantages against direct solver in memory cost. Thus, in conclusion, the ILDL solver is found to be the potentially good alternative of direct solver rather than an iterative solver. So, the expected reader would be the user of direct solver. However, their performance analysis is presented mainly for ILDL vs iterative ABF solver rather than vs direct solver, especially in the parallel performance section. I encourage the author to continue this work, but the presentation should be improved and more detailed performance analysis should be addressed before I recommend this for publication in SE.

We argue that these "extreme" cases are in fact those of greatest interest to many practitioners, in particular in geodynamics. It's become fairly common practice to abstract and characterize some of the difficulties of challenging heterogeneous coefficient structures with multiple-sinker problems, as they conveniently offer two "knobs" for contrast and geometrical complexity (number of sinkers). Each parameter tends to stress solvers in different ways.

We have attempted to stress in the paper (e.g. lines 257, 321, and 396) that in addition to the lower memory footprint, a perhaps even greater advantage of tools like ILDL preconditioning is their relative ease and robustness of use compared to an ABF solver.

The ABF solver presented here took years to tune, for experts. While they do indeed provide "optimal" solvers in many of the ways optimality is defined, they certainly have not shown themselves to be optimal in terms of solvers that most practitioners can understand and implement enough to incorporate. As stated above, the readers with the most to gain from ILDL preconditioning are those who rely on direct solvers, but who run into a memory limitation. Whereas before, the only good option was a risky exploration of solvers like ABF (or monolithic multigrid solvers), we aim to demonstrate the availability and performance of a far more incremental change to the solver stack.

It would have been tempting to not include the ABF solves at all - again, we agree that the most likely user of ILDL preconditioners would be those for whom this solver is practically out of reach, and we agree that the closer comparison is with direct solvers. However, we feel that it is important to clearly show what is attainable with a more invasive and complex implementation, particularly as we also believe strongly in the value of composable and hierarchical solver frameworks (such as the PETSc-based one used here) which offer a way forward to making such solvers easier to use, though there is clearly a long way to go.

The parallel section certainly leaves many things to future work. Our main objective, in analogy to existing work (and practice) which uses ILU subdomain solves within a block Jacobi or additive Schwarz preconditioner, is to simply demonstrate that a similar approach is available (with many of the same drawbacks) for indefinite problems, and the within a composable software environment, this is in many ways easier than one might expect to experiment with. Also see the response to point 17, below.

> Detail comments:
>
> 1. In introduction: Several sentences sound your opinion rather than the objective view (e.g. "This is unfortunate" in line 102). Such phrases are not appropriate for the research paper.

We agree that we veered too far into this territory, and have thus edited the introduction to remove several opinion-based statements, and have made others more objective.

> 2. In introduction: Please more review the progress and difficulty in direct solvers, although the author mainly reviews the recent progress of iterative solver.

While there is certainly still research going on, the field of direct solvers could be considered to be much more mature, and is extensive. As such we've added a comprehensive review article, Davis et al. 2016, for the interested reader. We also note that the references specifically on incomplete $LDL^T$ factorization can guide the interested reader to that subset of the direct solver literature which is most relevant (methods using complete $LDL^T$ factorizations).

> 3. In line 45: The hieratical grid system with such as AMR [Rudi et.al. 2015] worked well as the solution of highly variable viscosity problem with controlling the coefficient.

We note that the referenced work uses *smoothed* viscous inclusions in their benchmarks, so while addressing the problem of large viscosity contrasts, does not involve actual discontinuities in the coefficient field, so that coefficient always varies sharply across a single finite element - this is currently mentioned in the paper on line 277.

Nevertheless, AMR-based methods show great promise. However, on a practical level, AMR methods are very challenging to implement, perhaps even more so than ABF methods. In contrast, we believe that the approaches highlighted in this work offer a tool which promises to be more useful to the large class of users who may want to move beyond using a direct solver, but only have the time or resources to experiment with something like ILDL preconditioners, which correspond to the ILU preconditioners which practitioners have been successfully using, albeit only for the positive-definite problems for which they are designed, for many years.

> 4. In line 95: Since expected readers of this journal are not specialists in linear algebra, a more comprehensive review is needed. For example, how much memory was saved against direct solver with increasing/decreasing the time-to-solution in the past successful application?

While this paper attempts to give some detailed information on the algorithms being used, we ultimately hope that the experiments (and provided source code) will be intelligible to readers without much linear algebra background, in the comparisons they offer to direct solvers, especially.

Some of the included references contain some information (Hagemann and Schenk 2006, for example), but a key part of the contribution of this paper is to address a general lack of this sort of information in the literature. Indeed, there aren't as many past successful applications as we believe there should be, and this paper is one step to try to expose these solvers. We perform a study of the relevant tradeoffs in the context of a set of physical applications, as opposed to in the context of a set of fixed-sized matrices representing a wide array of applications, which are more typical in the published work on these solvers.

> 5. In line 211: Delete the space after "("

This sentence has been reworded in response to comments from another reviewer.

> 6. In line 253: The spectral analysis for scaled pressure mass matrix can cite [1]

This citation has been added.

> 7. In Numerical experiment: I think that the experiment starts with x=0. This problem setting is suitable for steady-state solution. But in practice, we solve the timestepping/nonlinear problems. Thus, it is interesting if ILDL largely outperforms the direct solver from the second step. The solution of the previous step will be a good initial gauss for reducing the iteration of ABF and ILDL.

A simple motivation for using zero initial guesses is that this represents a uniformly-available and in some sense "worst-case" setup, which would be interpretable in the context of the most readers' problems.

In the case of a nonlinear solve, one is typically computing an update step (e.g. in Newton's method). There is no obviously generally useful initial guess for the linear solves; for example, the previous step would not be useful, as the solver has just updated the approximate solution with what it considers to be a good multiple of this search direction.

For nonlinear problems solved at each timestep, it is common practice to use a previous solution as an initial condition, but this is an orthogonal issue to the one of initializing solutions to the linear solves which may be performed within the nonlinear solver.

In the case of providing starting guesses for linear, time-dependent problems, an initial guess might provide faster convergence if one were relying on an absolute convergence tolerance. However, this approach is sensitive to scaling of the equations and as such the definition of an absolute tolerance is problem-dependent and must be carefully chosen, if used at all. The results in this paper are intended to be robust to system scaling (and as such we do not emphasize scalings or units - for more see the response to Reviewer 1, which is another motivation for not considering absolute convergence tolerances here.

> 8. In line 264: In practice, direct solver is mainly used in 2D problems. Also, in memory capacity, the difference in maximum element size in 3D ($40^3$ for PARADISO $<$ $48^3$ for ILDL) seems to be trivial but that in 2D ($252^2 < 332^2$) is significant in scientific application. Then, the experiment in 2D should worth considering in SE.

We very much agree that 2D problems with large memory footprints are relevant in practice. We performed these experiments (and note that they can also be performed using our included, open source code), but did not believe that they would be as interesting to the reader as the 3D problems which we've chosen to devote manuscript space to. This was not unexpected, as direct solvers exhibit better scaling for 2D problems than 3D ones, in particular with regards to memory footprint. This implies that far fewer readers would benefit from a deeper look at the 2D case, as there are fewer gains to be made over a direct solver. Those readers who have run into the limits of direct solver performance for 2D problems would likely be better served by investigating the more complex alternatives: ABF solvers, monolithic multigrid solvers, and emerging nonlinear solvers based on pseudotransient continuation.

Also see Figure 1 in our response to Reviewer 1, for a 2D experiment which we removed from an earlier draft.

> 9. In line 317: It is confusing that ABF does not fail to converge in Figure 1. Why not plot the case with contrast = 10ˆ6 with 8 inclusions?

This has been clarified in the text (line 325, and see also our response to a similar concern from Reviewer 1).

The plot resulting from changing the viscosity contrast $10^4$ to $10^6$ would look very similar.

> 10. In line 318: Do we really need to solve the problem with over 10 inclusions in 32ˆ3? The accuracy of such a setting seems to be a useless solution in physics. In addition, to check the robustness, SINKER box test of [May and Moresi, 2008] is better than this setting.

It is true that at these grid resolutions, the inclusions may well be under-resolved, and if one were interested in physically-relevant flow fields for these problems, finer grids should be used. We note that in applications, particularly those like geodynamics wherein coefficient structure emerges and changes with time, solvers must be able to handle under-resolved cases gracefully.

We use the multiple-sinker problem not because of its direct physical relevance, but because it presents a very useful abstraction of difficult coefficient structures that can appear in practice.

The advantages of this benchmark over something like SINKER are

- Distribution of interface alignment is uniformly distributed, so effects of grid alignment are less of a concern

- By changing the number of sinkers, the geometric complexity of the coefficient structure is quantifiable and adjustable.

On this second point, it can be observed that the spectrum of the operator, on which the convergence of the solver depends critically (See Elman, Silvester, Wathen 2005, Chapter 5) has a wider range of eigenvalues (worse conditioning) when the viscosity contrast is increased, and less tightly-cluster eigenvalues when the number of inclusions is increased. Since conditioning and spectral clustering are (for symmetric systems) key predictors of convergence for Krylov methods, this provides a very useful benchmark.

The SINKER benchmark might be an interesting complement, because of the complexity of flow induced by the corners of the rectangular inclusion.

> 11. In Figure 1: Sample glyphs are difficult to see.

These glyphs are only intended to give an impression of the flow field, and note that the images are high resolution and can be zoomed in on in a PDF file.

> 12. In Figure 1: What is the message from the peak memory foot point? Why memory size in Table 2 is not enough?

Table 1 only covers a single coefficient structure. We include the memory plot in Figure 1 in order to show, in a quick-to-see way, that the memory behavior is insensitive to the coefficient structure and to make the scaling behavior (the slope of the curves) apparent, hopefully giving the reader a clear idea of the memory footprint gains available over a range of problems.

> 13. In Table 2: For a fair performance comparison, it should be noted that the number of iterations independent from the DOF for ABF.

Table 2 is concerned with the effect of the drop tolerance on the ILDL-preconditioned solve, but Table 1 includes iterations counts for the ABF solver. This independence of iteration count on problem size for the ABF preconditioned solves isn't explicitly noted elsewhere, as the scalability of the full solves implies this.

> 14. In line 320: Since your ABF is based on Jacobi smoother and Arnoldi type Krylov method, more smoothing iteration or avoiding rounding error of GMRES are promising to gain the convergence even with 10ˆ6 problem. It is interesting to see the performance of ABF with increasing the number of inner smoothing iterations to converge 10ˆ6 problem (I argue that such simple tuning is out of the expertise.). Whether such robust ABF can solve the 10ˆ6 problem faster than the ILDL method or not, is the matter for ILDL to be the alternative of ABF.

It is indeed true that ABF solvers can be tuned to deal with large viscosity contrasts by increasing the amount of computational work done by the inner multigrid solver. For a given problem, which solver offers the fastest time-to-solution can indeed depend on the amount of effort spent in tuning the solver. The main point we wish to make, though, is that ILDL-preconditioned solves are less sensitive to changes in the coefficient structure than any given ABF solver. It is our anecdotal observation that the performance of the ILDL-preconditioned solves tends to at least degrade incrementally with viscosity contrast, whereas the ABF solve can completely stagnate at a certain contrast, requiring the sorts of tunings mentioned.

> 15. In line 297: Please write Eqs. (5), (11), and the norm should be a consistent form.

We have changed the notation in the mentioned equations and added a note on line 305.

> 16. In line 353: Additive Schwarz Method (ASM) should be noted.

Fixed.

> 17. In "Using ILDL within a parallel preconditioner": Since ILDL is worth investigating as an alternative of direct solver PRADISO rather than ABF solver, the performance on SMP system (openMP) is more interesting than distributed memory parallelization (MPI). Please reconsider the way of presentation. Since ABF is inherently suitable for the distributed memory parallelization, Table 3 did not show any advantage of ILDL.

We agree that this is a very interesting future avenue, and is an ongoing project in terms of research and software development. There are a few references at the end of the paper (line 410) on recent work, which we more properly introduce as mentioned in the next comment. Implementations of ILDL preconditioners which function in shared memory parallel environments (e.g. OpenMP) or fine-grained parallel environments (e.g. on GPUs) show obvious promise in terms of reducing the time-to-solution of the ILDL-preconditioned solves presented here, by parallelizing operations (though not notably affecting algorithmic properties like number of iterations). Most importantly, we note that future parallel implementations will not notably impact memory usage, which we have emphasized as the key limiting resource for practitioners relying on direct solvers.

> 18. In lines in 400-404: These lines seem to be a jump in the context. Please introduce them in more detail if you want to address them. By the way, "incomplete LDL" should be ILDL

We have tried to reword the concluding statements (line 407 and onwards) on extensions to the algorithms to make the presentation flow better.

ILDL is a synonym for "incomplete $LDL^T$", but we have made this change.

[1] P. P. Grinevich and M. A. Olshanskii, An iterative method for the Stokes-type problem with variable viscosity, SIAM Journal on Scientific Computing, 31 (2009), pp. 3959– 3978